# The coupled oxygen and carbon dynamics in the subsurface waters of the Gulf and Lower St. Lawrence Estuary and implications for artificial oxygenation

William A. Nesbitt[1], Samuel W. Stevens[2,3], Alfonso O. Mucci[4], Lennart Gerke[5,6], Toste Tanhua[5], Gwénaëlle Chaillou[7], and Douglas W.R. Wallace[1]

[1]Department of Oceanography, Dalhousie University, Halifax, Nova Scotia, Canada
[2]Hakai Institute, Heriot Bay, British Columbia, Canada
[3]Department of Earth, Ocean and Atmospheric Sciences, The University of British Columbia, Vancouver, British Columbia, Canada
[4]GEOTOP and Department of Earth and Planetary Sciences, McGill University, Montreal, Quebec, Canada
[5]Marine Biogeochemistry Research Division, GEOMAR Helmholtz Centre for Ocean Research Kiel, Kiel, Germany
[6][C]Worthy, LLC, Boulder, CO, USA
[7]Québec Océan and Institut des sciences de la mer de Rimouski (ISMER), Université du Québec à Rimouski, Rimouski, Québec, Canada

*Correspondence to :* William A. Nesbitt (william.nesbitt@dal.ca)

**Abstract.** The Gulf and Lower St. Lawrence Estuary have experienced major environmental change over the past century, including the development of hypoxic bottom waters and their simultaneous warming and acidification. Here, we use biogeochemical observations collected during the 2021-2023 TReX project as well as historical data, integrated with a tracer-calibrated 1D Advection-Diffusion model with variable boundary conditions to represent dissolved oxygen (DO) and dissolved inorganic carbon (DIC) dynamics within the core of the oxygen minimum 25 zone (27.15-27.3 kg m$^{-3}$ isopycnals) of the Laurentian Channel. The rate of in-channel oxygen utilization in the deep layer was nearly invariant from 2003 to 2023 at 21.1 ± 2.5 µmol kg$^{-1}$ yr$^{-1}$ and the DIC accumulation rate was estimated to be 18.3 ± 2.5 µmol kg$^{-1}$ yr$^{-1}$. Using δ$^{13}$C$_{DIC}$ data, we assess the effect of microbial organic matter remineralization processes and dilution of the $^{13}$C$_{DIC}$ pool (−6.6×10$^{-3}$ ‰ per µmol of added metabolic DIC). This combination of long-term observations and tracer-informed transport modeling helps reconcile differences in prior 30 estimates of biogeochemical transformation rates and improves our ability to predict deepwater DO and DIC cycling. Finally, we apply the model to the mitigation scenario proposed by Wallace et al. (2023) for artificial re-oxygenation of the Laurentian Channel bottom waters using pure oxygen. We estimate that the injection of ~8.3 × 10$^5$ tonnes yr$^{-1}$ of oxygen, equivalent to an additional 55 µmol kg$^{-1}$ relative to the 2023 boundary concentration proximal to the Cabot Strait, would be required to achieve and maintain above hypoxic levels (>62.5 µmol kg$^{-1}$) at 35 the head of the Laurentian Channel. Using the model, we estimate the time required to re-establish steady-state along-channel distributions of DO and DIC following a change in offshore boundary conditions to be about 10 years, or twice the along-channel transit time. This study provides new long-term characterizations of deepwater DO and DIC cycling and offers a first-order assessment of the feasibility of large-scale re-oxygenation in the Gulf and Lower St. Lawrence Estuary.


# 1 Introduction

## 1.1 Background

Over the past century, the number of documented coastal hypoxic zones (Dissolved Oxygen (DO) < 62.5 μmol kg$^{-1}$) has swelled (Breitburg et al., 2018; Diaz and Rosenberg, 2008; Vaquer-Sunyer and Duarte, 2008). Historically, hypoxia has developed as a result of eutrophication, driven by elevated nutrient inputs from rivers and groundwater (Rabalais et al., 2014). However, climate-induced changes in circulation, increased stratification and weaker mixing events ("reduced ventilation") are, increasingly, exacerbating this issue (Helm et al., 2011; Jutras et al., 2023a; Oschlies et al., 2018). In many cases, hypoxic zones develop seasonally in coastal environments that experience seasonal stratification and periodic ventilation as a result of wintertime mixing (e.g., Gulf of Mexico, Chesapeake Bay, Bedford Basin; Hagy et al., 2004; Murphy et al., 2011; Rabalais et al., 2001, 2002; Rabalais and Turner, 2019; Rakshit et al., 2023; Testa et al., 2017). More persistent hypoxia, and even anoxia, occurs in permanently stratified estuaries and coastal seas such as the Gotland Basin in the Baltic Sea (Conley et al., 2009, 2011), the Black Sea (Capet et al., 2013; Murray et al., 1989, 1991), as well as the Gulf (GSL) and Lower St. Lawrence Estuary (LSLE) in Eastern Canada (Genovesi et al., 2011; Gilbert et al., 2005; Jutras et al., 2020, 2023b; see Fig. 1).

The LSLE and GSL is a notable region where the expansion and intensification of a persistent hypoxic zone, accompanied by subsurface warming (Galbraith et al., 2024; Thibodeau et al., 2010), has been well documented. The water column in the LSLE and GSL is characterized by strong vertical stratification, with a surface layer that flows seaward, a cold intermediate layer (CIL; 50–150 m; Galbraith, 2006) formed in winter in the GSL and which flows landward, and a landward-flowing deep layer (>150 m) that originates in the western North Atlantic. The development of hypoxic bottom waters has been attributed primarily to a change in the water-mass composition entering the GSL through the Cabot Strait, with additional contributions from enhanced organic matter (OM) export and remineralization at depth (Gilbert et al., 2005; Jutras et al., 2020, 2023b). The latter process not only consumes DO but also leads to the accumulation of metabolic $CO_2$, contributing to the dissolved inorganic carbon (DIC) inventory and acidification of the deep layer (Mucci et al., 2011). Beyond potential and documented (Pascal et al., 2024; Robert, 2022) negative impacts on marine biodiversity (e.g., habitat compression, alteration of predator prey dynamics, impacts on growth, mobility, and reproduction; Breitburg, 1992; Diaz and Rosenberg, 2008; Wu, 2002), hypoxia in the LSLE and GSL has altered biogeochemical cycles (Lefort et al., 2012), including the nitrogen cycle, with denitrification and nitrous oxide ($N_2O$) production occurring at higher DO concentrations than previously documented (Pascal et al., 2025). Recently, Wallace et al. (2023) proposed use of pure oxygen, a by-product from the rapidly developing green hydrogen industry, as a possible mitigation strategy to address the growing hypoxia problem in the LSLE and GSL.

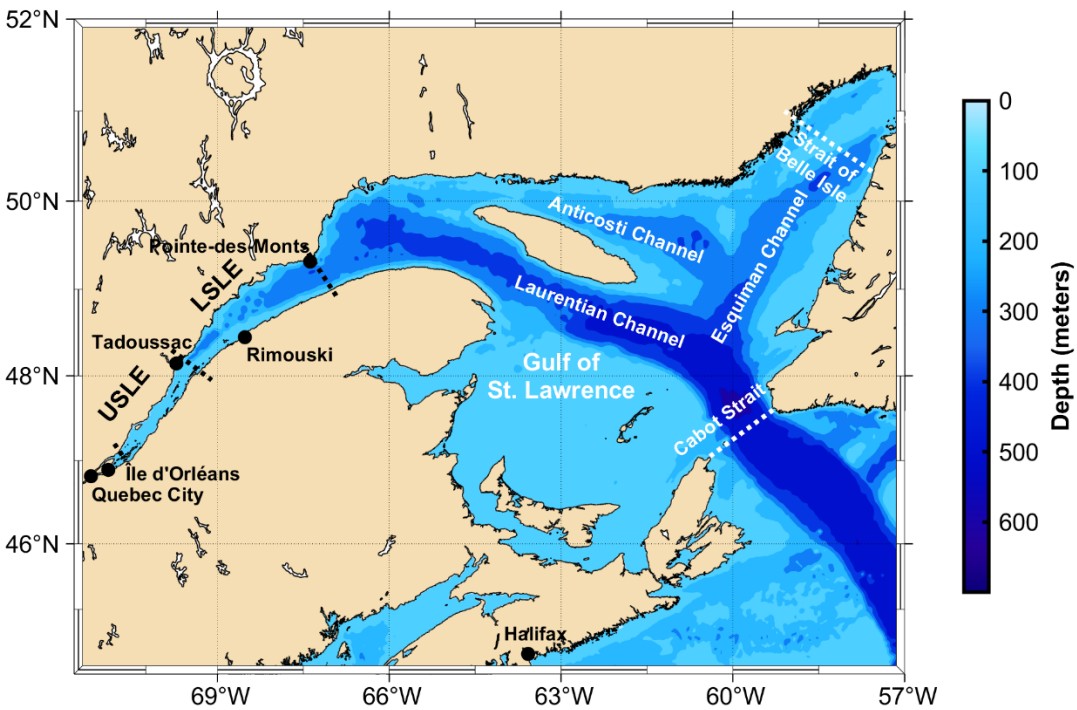

**Figure 1: Map and bathymetry of the Gulf and St. Lawrence Estuary. Each of the three main deep channels is labeled along with the two straits that connect with the western North Atlantic. Key landmarks are identified as well as the boundaries between the Upper St. Lawrence Estuary (USLE), Lower St. Lawrence Estuary (LSLE), and Gulf of St. Lawrence (GSL).**

## 1.2 Previous modeling studies

Advection-diffusion models of varying complexity, including 1D (Bourgault et al., 2012; Lehmann et al., 2009; Savenkoff et al., 1996), 2D (Benoit et al., 2006), and integrated numerical approaches (Jutras et al., 2020, 2023b), have been employed to describe the transport and transformation of dissolved tracers in the LSLE and GSL as well as provide a quantitative understanding of biogeochemical dynamics in the deep waters of the Laurentian Channel (LC; the main bathymetric feature of the GSL and LSLE). These models offer valuable insights into the physical and biological controls of tracer distributions within the LC, but their findings have been conflicting. Discrepancies arise from disagreements over the magnitude of key physical dispersion parameters, such as the advection velocity and diffusivity.

Previous estimates of along-channel advection velocities ($u$) range from $5 \times 10^{-3}$ to $10 \times 10^{-3}$ m s$^{-1}$ (Bugden, 1991; Gilbert, 2004; Rousseau et al., 2025) with horizontal eddy diffusivities ($K_H$) ranging from $2.8 \times 10^2$ to $8.2 \times 10^2$ m$^2$ s$^{-1}$ (Bugden, 1991; Savenkoff et al., 2001; see Table 1). Perhaps the most consequential, dissonant parameter in previous modeling efforts is the vertical eddy diffusivity ($K_Z$), with published estimates ranging over several orders of magnitude, from $1.0 \times 10^{-5}$ to $2.2 \times 10^{-3}$ m$^2$ s$^{-1}$ (Benoit et al., 2006; Bugden, 1991; Mertz and Gratton, 1995). Likewise, the oxygen utilization rate (OUR) within the inflowing deep layer has been estimated by a variety of

quantitative methods and models (Benoit et al., 2006; Bourgault et al., 2012; Jutras et al., 2020; Lehmann et al., 2009; Savenkoff et al., 1996) with only one attempt (Savenkoff et al., 1996) to include an estimate of the accompanying DIC accumulation rate. Previous OUR estimates (converted to $\mu$mol kg$^{-1}$ yr$^{-1}$) are summarized in Table 1 and a detailed summary of these studies can be found in the Supplemental Material (S1).

**Table 1: Previous estimates of the oxygen utilization rate (OUR; $\mu$mol kg$^{-1}$ yr$^{-1}$), estimated sediment oxygen demand (SOD; % of net OUR), horizontal advection velocity ($u$; m s$^{-1}$), horizontal diffusivity ($K_H$; m$^2$ s$^{-1}$), and vertical diffusivity ($K_Z$; m$^2$ s$^{-1}$) within the deep layer (<150 m) of the Laurentian Channel. When available, applied dispersion parameters are listed in adjacent columns.**

| Source | Method | OUR | SOD | $u$ | $K_H$ | $K_Z$ |
|---|---|---|---|---|---|---|
| Savenkoff et al. (1996) | 1D Adv-Diff | 14.3 | 60 | $5.0 \times 10^{-3}$ | | |
| Benoit et al. (2006) | 2D Adv-Diff | 8.3-39.8 | 100 | $1.0 \times 10^{-2}$ | $8.2 \times 10^2$ | $8.2 \times 10^{-4}$ |
| Lehmann et al. (2009) | 1D Diff-React | 53.4 | 64 | $1.0 \times 10^{-2}$ | | |
| Bourgault et al. (2012) | 1D Adv-Diff | 24.2 | 36 | $5.0 \times 10^{-2}$ | | $5.0 \times 10^{-5}$ |
| | | 48.7 | 17 | $1.0 \times 10^{-2}$ | | $1.0 \times 10^{-4}$ |

### 1.3 Research objectives

Recently, based on results of tracer measurements (CF$_3$SF$_5$) carried out during the Gulf of St. Lawrence Tracer Release Experiment (TReX), Stevens et al. (2024) reported an average along-channel advection speed ($u = 5.0 \times 10^{-3}$ m s$^{-1}$) and diffusivities ($K_H = 2.0 \times 10^2$ m$^2$ s$^{-1}$, $K_Z = 1.0 \times 10^{-5}$ m$^2$ s$^{-1}$). While previous estimates of $K_Z$ in the region have been obtained using microstructure profiling (Cyr et al., 2011), these TReX parameters represent the first set of dispersion coefficients derived from tracer measurements in this system, making this an ideal opportunity to revisit estimates of OM remineralization rates within the LC.

In this study, we adapt the tracer-calibrated 1D advection-diffusion model of Stevens et al. (2024) to re-evaluate coupled DO-DIC dynamics within the deep layer of the LC and provide new estimates of OUR, DIC accumulation rate, and dilution of the $^{13}$C$_{DIC}$ pool, as well as identify their respective sources/sinks. Historical data sets are utilized to complement the large new data set collected during TReX to evaluate variations of the deep layer OUR rates over the past two decades (2003-2023). The carbon source for in-channel DIC accumulation is also identified on the basis of its stable carbon isotopic signature ($\delta^{13}$C$_{DIC}$). Finally, the time required to reach a new steady-state (constant channel-end tracer concentration) following a change in boundary conditions ([DO] of bottom waters near Cabot Strait) is estimated together with implications for recently proposed mitigation efforts. In summary, this paper provides an updated estimate of average deep-layer remineralization rates, identifies the source of bottom-water DIC, and provides a first-approximation prognosis of anticipated re-oxygenation efforts.

## 2 Methodology

### 2.1 The Gulf and St. Lawrence Estuary

The Gulf (GSL) and St. Lawrence Estuary (SLE) comprise the largest semi-enclosed estuarine system in the world, linking the Great Lakes to the North Atlantic Ocean via the St. Lawrence River, its estuary, the GSL as well as the Cabot Strait and Strait of Belle Isle (see Fig. 1). With a drainage basin of 1.32 million km$^2$, the St. Lawrence River channels the second largest freshwater discharge (11,900 m$^3$ s$^{-1}$) on the North American continent, second only to that of the Mississippi. The SLE begins at the eastern tip of Île d'Orléans (10-15 km east of Québec City) and

stretches 400 km seaward to Pointe-des-Monts where it widens into the GSL. The SLE is typically divided into two segments based on its bathymetry and hydrographical features, the Upper St. Lawrence Estuary, and the LSLE. The LSLE extends from Tadoussac to Pointe-des-Monts, is 30 to 50 km wide and <300 m deep, with a smooth bottom topography and strong water column stratification. The LSLE opens into the GSL, a substantial semi-enclosed sea connected to the North Atlantic Ocean through the Cabot Strait to the south and the Strait of Belle Isle to the north.

The Laurentian Channel (LC), or Trough, is the main bathymetric feature of the GSL and LSLE and is a deep (250-600 m) U-shaped valley (10-100 km wide) that extends 1240 km from the North American continental shelf break to the head of the LSLE near Tadoussac (see Fig. 1). Two other, shorter channels branch off the LC within the GSL: the Anticosti Channel branches to the west, north of Anticosti Island, whereas the Esquiman Channel extends northeastward towards the Strait of Belle Isle.

Historically, two distinct water masses contributed to the inflowing deep layer through the Cabot Strait: the warm (4.4 to 8.0 ºC), saline (S$_P$ = 35.0 to 35.2, where S$_P$ stands for practical salinity), oxygen-poor (155 to 250 µmol kg$^{-1}$) North Atlantic Central Water (NACW) and the colder (-0.7 to 3.2 ºC), fresher (S$_P$ = 33.4 to 35.0), oxygen-rich (280 to 310 µmol kg$^{-1}$) Labrador Current Water (LCW; Gilbert et al., 2005; Jutras et al., 2020). Over the past century, the deep waters of the LC have experienced significant changes, including deoxygenation (Gilbert et al., 2005; Jutras et

al., 2020, 2023b), warming (Galbraith et al., 2024; Thibodeau et al., 2010),  and acidification (from metabolic CO$_2$ accumulation; Mucci et al., 2011). It is estimated that, between the early 1930s and 2003, one-half to two thirds of the decline of DO concentrations but all the temperature increase in the deep layer of the LSLE and GSL was due to a reduction of the LCW contribution to the deep layer (and concomitant increase in NACW; Gilbert et al., 2005), with further DO depletion and acidification brought on by the increased degradation of OM exported to depth due to

increased delivery of nutrients from the St. Lawrence River (Genovesi et al., 2011; Gilbert et al., 2005; Jutras et al., 2020, 2023b; Mucci et al., 2011; Thibodeau et al., 2006, 2010).

From the late 1990s to 2008, water mass mixing ratios and, therefore, the input of oxygen from the Atlantic Ocean via Cabot Strait were nearly constant (Jutras et al., 2020). From 2008-2018, however, the relative contribution of the LCW steadily decreased from ~40% to ~20% with a corresponding reduction in the DO concentration (from 165

150  µmol kg$^{-1}$ to 125 µmol kg$^{-1}$) in the inflowing deep layer (Jutras et al., 2020). As of 2020, the relative contribution of LCW to water entering the LC at Cabot Strait was nearly null and, thus, inflowing deep waters was composed almost entirely of NACW (Jutras et al., 2023b). Consequently, the minimum DO concentrations in the LSLE have dropped to ~30 µmol kg$^{-1}$ or less than 10% saturation. The decreasing contribution of the LCW to the deep waters of

the GSL is the result of a change of the ocean circulation in the western North Atlantic, more specifically a retroflection of the Labrador Current off the tip of the Grand Banks of Newfoundland, a shift that is partly driven by a northward shift of the wind patterns in the western north Atlantic (Jutras et al., 2023a).

## 2.2 Sampling

Water samples were collected onboard the R/V Coriolis II throughout the LSLE and GSL from October 2021 to June 2023 over the course of four separate research cruises conducted as part of the TReX Project. One of the objectives of TReX was to obtain a comprehensive view of biogeochemical fluxes while simultaneously measuring the tracer distribution in the LSLE and GSL. Two of these cruises were conducted in partnership with existing programs and missions: the Fall 2022 cruise was carried out in collaboration with the Department of Fisheries and Oceans' Atlantic Zone Monitoring Program (AZMP; Blais et al., 2023), and the Summer 2023 cruise was conducted jointly with the RQM-led PLAINE mission. Water samples were taken at specified depths using 12 x 12 L Niskin PVC bottles mounted on a rosette outfitted with a SeaBird SBE 911 Conductivity-Temperature-Depth (CTD) probe and an SVE-43 DO probe. The factory calibrations of both the conductivity and DO probes were verified and corrected post-cruise following the collection and analysis of discrete Niskin-bottle samples for practical salinity ($S_P$; conductivity measurements using a Model 8410A Portasal salinometer) and DO (Winkler titrations; Grasshoff et al., 2009). Samples taken for DIC-$\delta^{13}C_{DIC}$ and total alkalinity ($A_T$) were transferred directly from the Niskin bottles with a gas-tight tube into 250 mL borosilicate glass bottles, rinsing the bottle twice with the sample water before overflowing by one full volume while avoiding trapping bubbles, and leaving no headspace. Samples were immediately poisoned with 0.05 mL of a saturated $HgCl_2$ solution, before being capped with plastic screw tops. Samples were stored at room temperature before being transported to the CERC.OCEAN Laboratory at Dalhousie University for analysis, typically performed within less than 3 months. Maps of sampling locations for all years can be found in the Supplemental Material (S2 & S3). The TReX data were supplemented with historical data (depth, T, $S_P$, [DO]) provided by Alfonso Mucci at McGill University. His analytical methods are described in detail in the Supplementary Material (S3).

## 2.3 Laboratory analyses

DIC and its stable isotopic composition, $\delta^{13}C_{DIC}$, were measured simultaneously on discrete samples using Cavity Ring-Down Spectroscopy. This method utilizes an Apollo SciTech AS-D1 acidification system coupled to a Picarro G2201-I detector. A detailed description of the procedure can be found in Cheng et al. (2019) and Su et al. (2019). Calibration of the DIC analyses was conducted using certified reference materials (CRM Batch #'s 186 & 197; provided by A.G. Dickson, Scripps Institute of Oceanography, La Jolla, California, USA) whereas three, in-house reference materials (Baking Soda, $Na_2CO_3$, and $NaHCO_3$) were used to calibrate for $\delta^{13}C_{DIC}$ ($\delta^{13}C$ = -2.55, -10.51, and -20.89 ‰). The reproducibility of the DIC and $\delta^{13}C_{DIC}$ measurements was, respectively, better than $\pm$ 4 µmol kg$^{-1}$ and $\pm$ 0.15 ‰. The $\delta^{13}C_{DIC}$ is expressed in per mil deviations from the standard reference material Vienna-PeeDee Belemnite (VPDB) (Eqn. 1) on a normalized scale to NBS19 calcium carbonate (+1.95 ‰) and L-SVEC lithium carbonate (-46.6 ‰). According to Cheng et al. (2019), cross-laboratory reproducibility is $\pm$ 0.06 ‰ when applying

consensus-based corrections, highlighting the potential for high accuracy in line with the ±0.05 ‰ objective set by
the Global Ocean Observing System (GOOS).

$$\delta^{13}C_{DIC} = \left(\frac{(^{13}C/^{12}C)_{DIC}}{(^{13}C/^{12}C)_{V-PDB}} - 1\right) * 1000 \tag{1}$$

$A_T$ was determined by automated potentiometric titration using an in-house prepared 0.1M HCl titrant with a
Dickson Total Alkalinity Titrator composed of a Metrohm Dosimat 876 Plus dosing system, an Agilent 34970A
voltmeter, and a Metrohm LL-Ecotrode Plus pH electrode (Dickson et al., 2007). Calibration curves for $A_T$ were
constructed using in-house reference materials (gravimetrically prepared $Na_2CO_3$, solutions) that had been compared
against certified reference materials (CRM Batch #'s 186 & 197; provided by A.G. Dickson, Scripps Institute of
Oceanography, La Jolla, California, USA). The reproducibility of the $A_T$ measurements, based on replicate
measurements, was better than ± 3 μmol kg$^{-1}$.

**2.4 1D advection diffusion model**

We employ a 1D advection-diffusion model to derive a first-order estimate of the oxygen utilization rate (OUR),
DIC accumulation rate in the water column, and $^{13}C_{DIC}$ dilution rates in the core of the deep layer inflow of the LC
(i.e., the 27.15-27.30 kg m$^{-3}$ density surface or between 200-350 m with a mean depth of 270 m). The transport and
reaction of these constituents is represented as:

$$\frac{\delta c}{\delta t} + u\frac{\delta c}{\delta x} = K_H \frac{\delta^2 c}{\delta x^2} + S \tag{2}$$

where c represents the depth-averaged tracer concentration, x is the along-channel distance, t is time, u = 5.0 × 10$^{-3}$
m s$^{-1}$ is the mean advection velocity, $K_H$ = 2.0 × 10$^2$ m$^2$ s$^{-1}$ is the horizontal diffusivity, and $S$ is a constant first-order
source or loss term (OUR, DIC accumulation, $^{13}C_{DIC}$ dilution) representing the cumulative effects of biological
metabolism. The advection and diffusion parameters are assumed to remain constant throughout the model domain
and were estimated from the measured dispersion of an inert tracer (CF$_3$SF$_5$) introduced during the TReX Deep
experiment (Stevens et al., 2024).

We employ a finite difference approach to solve the advection-diffusion equation using an upwind scheme for the
advection term and a centered difference scheme for the diffusion term. The model domain extends along the length
of the LC from the Cabot Strait ($x = 0$ km) to the head of the LSLE and LC at Tadoussac ($x = 800$ km) and is
discretized into 100 grid points with a uniform spatial step size of 8 km. The model was also run with a spacing of
1000 grid points to check the impact of increasing the resolution and results were identical within the uncertainty.
To simulate changing DO concentrations in the inflow waters, we vary $c(0, t)$ over time. Estimates of the inflowing
DO concentrations between the 27.15-27.3 kg m$^{-3}$ density surfaces from 1999 to 2020 were taken as the mean of
measurements close to Cabot Strait (Mathilde Jutras, pers. comm., 2024; see Fig. 2). Inflowing DO concentrations
from 2021 onwards are estimated from the intercept of the along-channel DO concentration gradients measured
between the above-mentioned density surfaces. Since the estimated transit time for the 800 km transect is ~5.1 years
(Stevens et al., 2024) and the model is run between 2003 and 2023, every model run is initiated in 1999 to

encompass changing water properties for at least one full transect. At $x = 800$ km, a Neumann boundary condition is applied with a zero-gradient approximation, effectively enforcing no flux across the boundary, a realistic assumption given the steep bathymetry at the head of the channel.

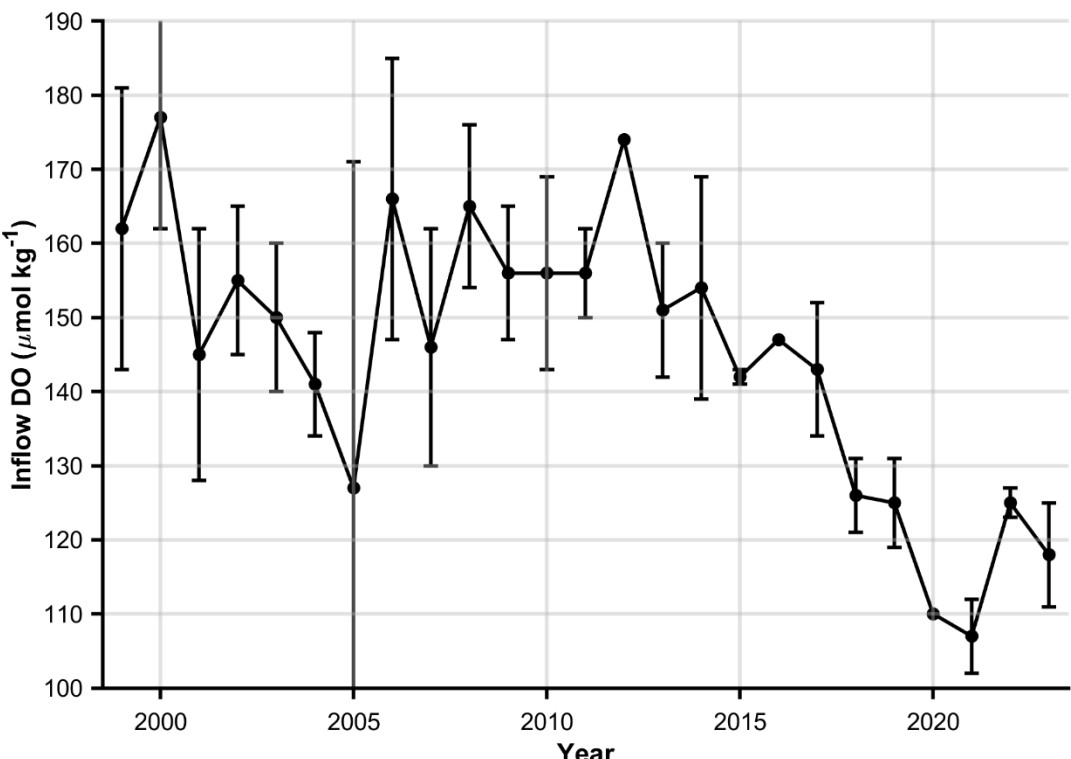

**Figure 2: Time series of dissolved oxygen (DO) concentrations (μmol kg⁻¹) in the deep-layer inflow through the Cabot Strait, defined between the 27.15–27.3 kg m⁻³ isopycnal surfaces. Values from 1999–2020 were provided by Mathilde Jutras (pers. comm.). Estimates from 2021 onward were derived from the intercept of the along-channel DO gradient.**

To estimate OUR within the deep-layer inflow of the LC, we vary the utilization term $S$ over a reasonable range of

values to produce estimates of $c(x, t)$ between 2003 and 2023. We then use a least-squares approach to minimize the misfit between modeled and observed DO for every year where observations are available, providing us with annual estimates of $c(x, t)$ and OUR. The initial range of $S$ values was estimated by dividing the mean along-channel DO gradient by the approximate transit time of 5.1 years over the 800 km study region (Bugden, 1991; Stevens et al., 2024), and scaling by the variance in observed gradients over the study period. Each model simulation is initiated in

1999 and runs continuously to the specific sampling year to account for interannual variations in the boundary condition at $x = 0$ and its cumulative influence on measured DO along the channel. Finally, we utilize the same approach to estimate the DIC accumulation and $^{13}C_{DIC}$ depletion rates from the 2021-2023 DIC-$\delta^{13}C_{DIC}$ dataset. The inflow boundary conditions for these estimates are determined using along-channel linear regressions extrapolated to the inflow location ($x = 0$, see Supplemental Material S4). The boundary conditions, least-squares fit sources terms,

and calculated yearly rates are presented in tables in Supplemental Material S5.

## 3 Results and discussion

### 3.1 Observations

The observed distributions of DO, DIC, and $\delta^{13}C_{DIC}$ exhibit clear horizontal gradients along the 27.25 kg m$^{-3}$ isopycnal surface (see Fig. 3). This density surface coincides with the depth of locus of the DO minimum (Gilbert et al., 2005; Jutras et al., 2020, 2023b; Mucci et al., 2011). At all locations and throughout the entire water column, DO and $\delta^{13}C_{DIC}$ decrease with depth until the DO minimum layer, whereas, in contrast, DIC increases. Within the deep layer and during the TReX project (2021-2023), DO concentrations decrease landward by ~110 μmol kg$^{-1}$ between the Cabot Strait and Tadoussac at the head of the LC/LSLE (Fig. 3a). In comparison, DIC increases by ~90 μmol kg$^{-1}$ along the same transect and its stable carbon isotopic composition (Fig. 3b), $\delta^{13}C_{DIC}$, decreases by ~1‰ (see Fig. 3c). These observations suggest that significant remineralization occurs within the deep layer, generating a strong gradient of metabolites (DO, DIC) along the deep-water flow path from the Atlantic Ocean westward towards the head of the LSLE.

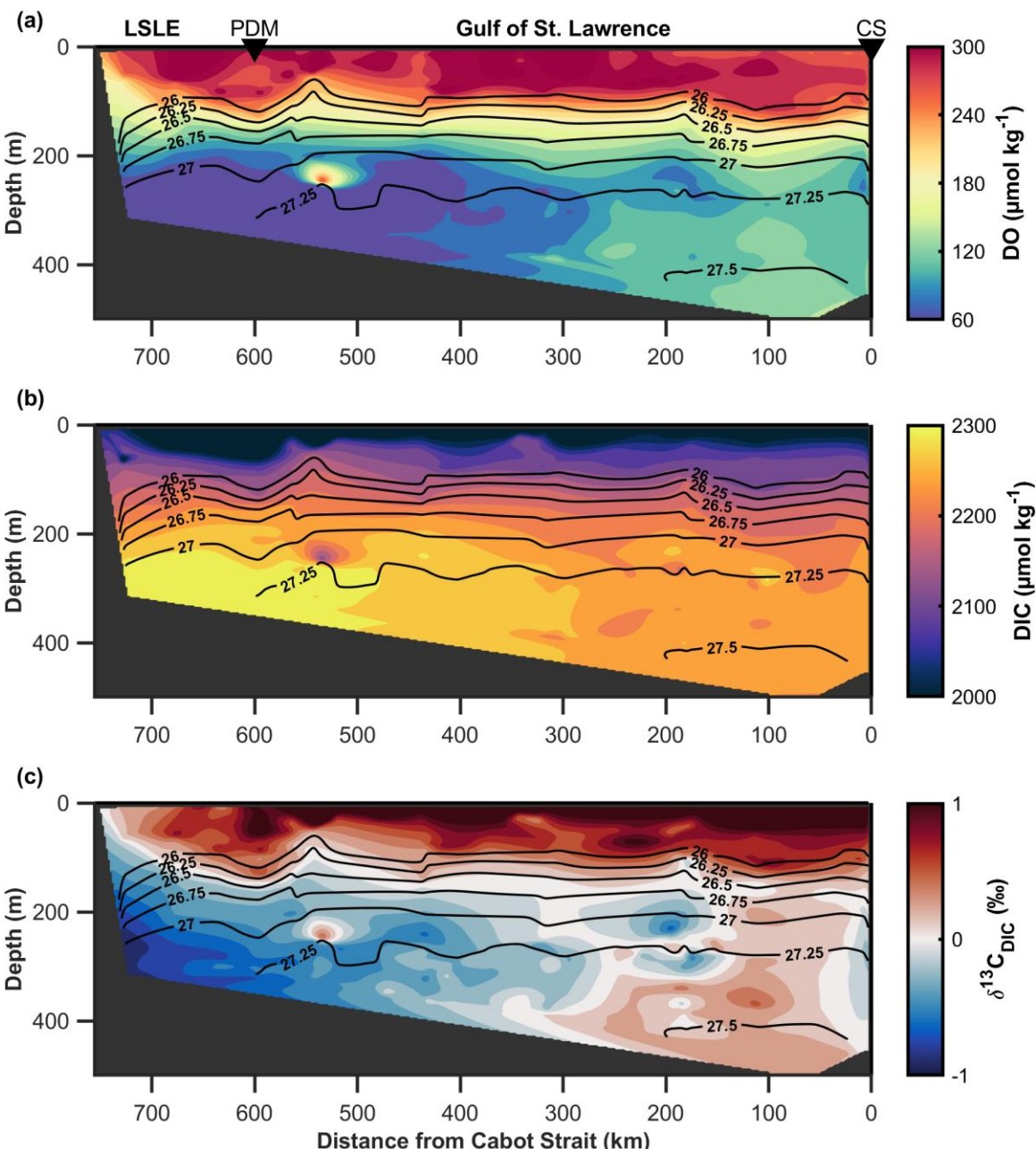

**Figure 3: Along-channel sections of (a) dissolved oxygen (DO) and (b) dissolved inorganic carbon (DIC) concentrations, (c) $\delta^{13}C_{DIC}$ in the Laurentian Channel (LC). The data were collected over the course of four cruises during the 2021–2023 TReX project. Discrete data from each individual survey were interpolated onto a grid using a natural neighbor interpretation for DO, DIC, and $\delta^{13}C_{DIC}$ and linear interpolation for isopycnals. The x-axis shows the distance (in km) from the Cabot Strait (CS), the entry point for the deep-layer inflow into the Gulf of St. Lawrence. Key locations are marked above sub-plot A with black triangles: Cabot Strait (CS), Pointe-des-Monts (PDM), that define the transition between the Gulf of St. Lawrence (GSL) and the Lower St. Lawrence Estuary (LSLE) (Fig. 1). Contoured black lines denote the mean isopycnal distribution, with key density surfaces labeled.**

### 3.2 One-dimensional advection-diffusion modelling

Recent tracer observations reveal that the local timescale for average horizontal transport within the deep layer of the LC is far less than that of vertical mixing (Stevens et al., 2024). The timescale for vertical eddy diffusion can be estimated from dimensional analysis (Bourgault et al., 2012) as follows:

$$\tau_d \equiv \frac{H_b^2}{K_z} \tag{3}$$

where $\tau_d$ is the diffusive timescale, $H_b$ is the vertical thickness of the ventilating layer, and $K_z$ is the vertical
diffusivity (Bourgault et al., 2012). This relationship yields a layer-average vertical mixing timescale of $\tau_d \approx 30$ years for a $H_b = 100$ m deep layer with $K_z = 1.0 \times 10^{-5}$ m$^2$ s$^{-1}$, as opposed to an average horizontal advection timescale of ~5 years. The large difference between the vertical and horizontal transport timescales justifies the use of a 1D advection-diffusion model to describe the DO-DIC dynamics in the deep layer of the LC.

The tracer-based estimates of Stevens et al. (2024) report basin-wide vertical mixing with effective $K_z$ on the order
of $10^{-5}$ m$^2$ s$^{-1}$. These estimates include larger localized diffusivities near boundaries (i.e. slopes and topographic features) of ~$1.5 \times 10^{-5}$ m$^2$ s$^{-1}$, but lower values in the channel interior of ~$6.5 \times 10^{-6}$ m$^2$ s$^{-1}$, and are of the same order of magnitude as previous microstructure measurements taken in the LSLE (Cyr et al., 2011). Nonetheless, localized enhancements in vertical mixing may contribute to the spatial heterogeneity in DO concentrations proximal to slopes and topographic features. To assess the influence that vertical oxygen supply to the deep layer
may incur, we estimated vertical diffusive fluxes using Fick's Law.

$$F = -K_z \frac{\partial C}{\partial z} \tag{4}$$

where $K_z$ is the vertical diffusivity, and $\frac{\partial C}{\partial z}$ is the vertical gradient in DO concentration across the diffusive depth (in this case 100 m) of the Cold Intermediate Layer (CIL) and the deep layer. Using data from 2022 (a year with dense observational coverage), we define the concentration gradient ($\partial C$) as the difference between the average CIL DO
concentration (~254 μmol kg$^{-1}$) and the assumed inflow DO concentration at Cabot Strait (125 μmol kg$^{-1}$), effectively removing the influence of in-channel respiration. For consistency with our dimensional analysis, we use a vertical diffusive distance ($\partial z$) of 100 m. We evaluate three plausible scenarios for $K_z$: (1) a basin-wide average of $1.0 \times 10^{-5}$ m$^2$ s$^{-1}$ (Stevens et al., 2024), (2) an elevated boundary-layer value of $1.5 \times 10^{-5}$ m$^2$ s$^{-1}$ (Cyr et al., 2011; Stevens et al., 2024), and (3) a hypothetical hotspot value of $1.0 \times 10^{-4}$ m$^2$ s$^{-1}$. These scenarios yield vertical DO
fluxes into the deep layer of approximately 1.1, 1.7, and 11.1 μmol kg$^{-1}$ yr$^{-1}$, respectively. When compared to the average modelled oxygen utilization rate (OUR $\approx 21.1$ μmol kg$^{-1}$ yr$^{-1}$), these correspond to ~5%, ~8%, and ~53% of the value of the OUR, respectively. These estimates indicate that vertical diffusion is a secondary contributor to the DO budget of the deep layer on a regional scale. Even in the hypothetical hotspot scenario, the elevated flux is spatially limited and unlikely to significantly contribute to the regional DO budget. It should be noted that localized

enhanced mixing would not only contribute to vertical diffusive supply of DO but may also produce horizontal buoyancy gradients and associated cross-channel circulation (e,g, Tang, 1983), redistributing oxygen laterally. These processes are not explicitly captured in our 1D model and would require more sophisticated modelling approaches to quantify.

To test the sensitivity of the modelled DO and DIC distributions to changes of the horizontal transport parameters $u$ and $K_H$, we conduct a simulation with a constant OUR of 20 µmol kg$^{-1}$ yr$^{-1}$ and fixed initial DO concentration of 125 µmol kg$^{-1}$ for one complete transit time of 5.1 years. The DO utilization estimates (cumulative amount of DO consumed during transport) at the head of the LC, i.e., at $x = 800$ km, calculated using various combinations of $u$ and $K_H$ are shown in Fig. 4. The value of $K_H$ appears to only have a noticeable impact on the end-of-channel DO utilization for very low advection speeds of $<2.0 \times 10^{-3}$ m s$^{-1}$. Hence, our model-based estimates are relatively insensitive to $K_H$ and we use the physical parameters estimated in Stevens et al. (2024), $u = 5.0 \times 10^{-3}$ m s$^{-1}$ and $K_H = 2.0 \times 10^2$ m$^2$ s$^{-1}$, for the simulations conducted in this study.

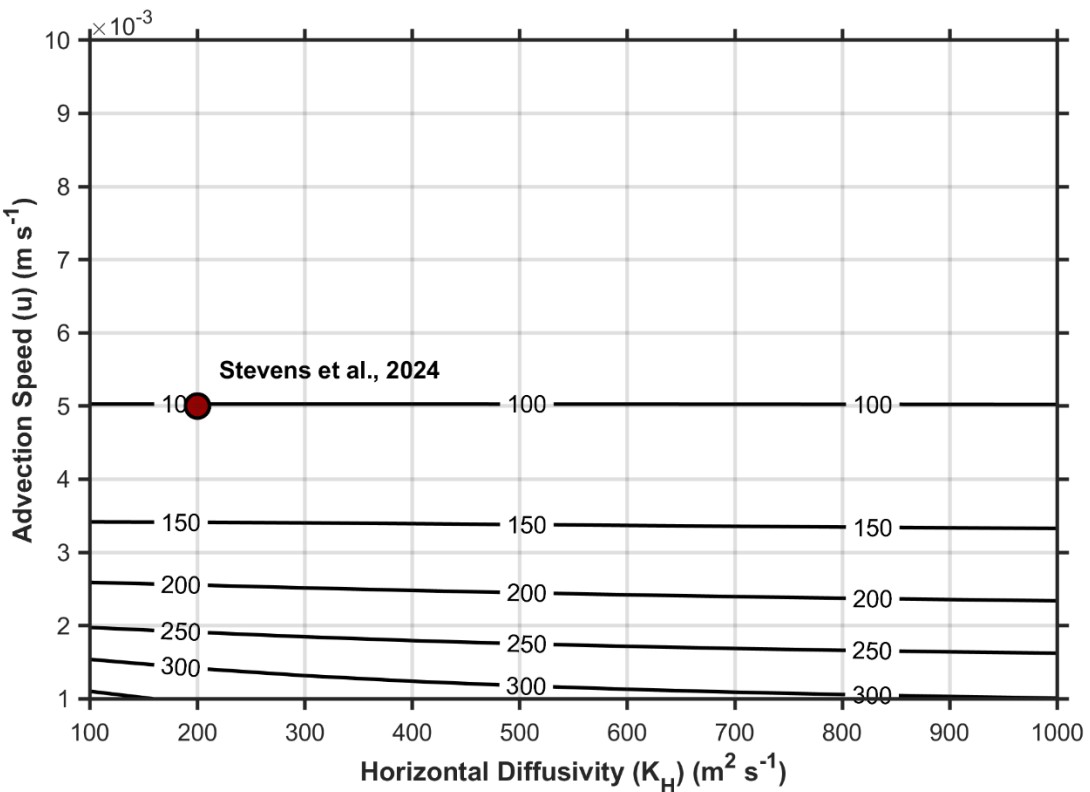

Figure 4: Sensitivity of the modeled DO concentrations at the head of the LC, i.e., $x = 800$ km with a fixed inflow concentration at $x = 0$ of 125 µmol kg$^{-1}$. Contours are the modeled end-of-channel ($x = 800$ km) DO utilization (amount of DO consumed during transport; µmol kg$^{-1}$) for a given pairing of $u$ and $K_H$. The combination of $u$ and $K_H$ from Stevens et al. (2024) that we later employ in this study is identified by a red marker on the figure.

### 3.3 Modeled DO results

Using the least-squares fitting method described above, the 1D advection-diffusion model is optimized against observed DO concentrations to determine a 'best-fit' OUR for the layer between the 27.15 and 27.3 kg m$^{-3}$ isopycnals (~200-350 m depth). The mean OUR over the 20-year period is 21.1 ± 2.5 µmol kg$^{-1}$ yr$^{-1}$, where the error represents the standard deviation (STDV) of the mean, giving a coefficient of variation (CV) of 12 %. Over this 20-year period, cumulative oxygen depletion is estimated at 107 ± 13 µmol kg$^{-1}$ during a full transect along the 800-km channel. These findings are consistent with the estimates of Jutras et al. (2020) and Bourgault et al. (2012; see Table 1), assuming that $u = 5.0 \times 10^{-3}$ m s$^{-1}$. The model output in Fig. 5 illustrates the spatiotemporal distribution of DO concentrations along the LC. Although observational data are not available for every year (missing years are 2004, 2005, 2008, 2012, and 2015) of the modelled period, boundary conditions exist for all years (see Fig. 2). For consistency, in the generation of Fig. 5, we utilize the time-mean OUR (21.1 ± 2.5 µmol kg$^{-1}$ yr$^{-1}$) derived from years with observational constraints, with the understanding that the simulations for the years without observations may carry a greater uncertainty. The modeled DO distributions presented in Fig. 5 accurately captures the gradual DO depletion and expansion of the hypoxic zone along the LC, with DO levels below 50 µmol kg$^{-1}$ near Tadoussac in recent years. Whereas the model does not capture spatial and seasonal variability, its representation of the along-channel oxygen gradient and cumulative depletion is within the expected range of field measurements. For example, it is known that the origin and reactivity of the settling OM varies along the estuarine gradient. Benoit et al. (2006) and Wang et al. (2025) reported that sediments (and, presumably deep waters) of the Upper St. Lawrence Estuary receive more allochthonous OM, whereas autochthonous OM dominates in the LSLE. This is consistent with the inferred (remote sensing datasets between 1998-2019) spatial and seasonal distribution of Chl α (in the Estuary and Gulf), including the presence of a mid-estuary maxima (Laliberté and Larouche, 2023), and would suggest a spatially varying remineralization potential and OUR. To evaluate the impact of a spatially heterogeneous OUR, we modified the 1D advection-diffusion model and doubled the remineralization rate over the final 200 km of the LC (approximately corresponding to the LSLE), while retaining the best-fit value elsewhere. This scenario was intended to represent enhanced respiration in nearshore regions due to elevated OM supply. The simulation was run to steady state (over a 10-year period or roughly 2 transit times) and used 2022 as a test year due to its extensive spatial data coverage. Whereas the reduction in least-squares misfit is modest (–2.3 %), the fit appears visibly improved in the LSLE, indicating a slightly better spatial alignment between model and observations (see Supplemental Material S8). This suggests that, although spatial heterogeneity in remineralization rates likely exists, the large-scale DO trends are effectively captured by a spatially averaged OUR in the present model configuration that integrates over regional scales.

The OUR estimated for the deep layer of the LC in this study falls within the range of previous estimates but remains at the lower end of other suboxic marine systems. For example, the Bedford Basin, a seasonally hypoxic and eutrophic fjord in Nova Scotia, exhibits a higher mean OUR of 413 ± 278 µmol kg$^{-1}$ yr$^{-1}$ (Rakshit et al., 2023), nearly an order of magnitude greater than the OUR in the LC deep layer. Additionally, sediment oxygen demand (SOD) utilizes an additional 37–76 µmol kg$^{-1}$ yr$^{-1}$ (Rakshit et al., 2023). Globally, the OUR in the LC sits at the

lower end of the observed range for pelagic respiration (across marine environments (6.5 to 24,400 μmol kg⁻¹ yr⁻¹; Robinson and Williams, 2005).

We assess the influence of mean annual temperature variations on the interannual modeled OUR estimates within the 27.15–27.3 kg m⁻³ isopycnal layer by applying both a linear regression and $Q_{10}$-based approximations to the model results. For each year, the modelled OUR value is paired with the mean temperature within this density range, ensuring that the analysis reflects year-specific conditions. The linear regression analysis reveals that temperature accounts for approximately 17% of the interannual variability of the OUR estimates ($R^2 = 0.171$), consistent with findings of Jutras et al. (2020, 2023b) who estimated that changing bottom-water temperatures contribute ~20% of the variation in deep-layer OUR. To further evaluate the temperature dependence of microbial respiration, we apply a $Q_{10}$ estimation that describes the temperature sensitivity of biological processes (microbial respiration). The $Q_{10}$ relationship follows:

$$OUR_T = OUR_{T_0} * Q_{10}^{(T-T_0)/10}$$

(5)

where $OUR_T$ represents the OUR for a given year, T is the mean annual temperature within the 27.15–27.3 kg m⁻³ isopycnal layer for that year, $OUR_{T_0}$ is the reference OUR at temperature T₀, and $Q_{10}$ quantifies the factor by which respiration rates change per 10°C increase. Based on previous studies and observed bottom-water temperatures, we select T₀ = 4.5°C as a representative long-term mean (Jutras et al., 2023b). The resulting best-fit $Q_{10}$ value of 0.33 is significantly lower than commonly reported values for marine respiration (~1.4–3.3; Genovesi et al., 2011; Robinson and Williams, 2005). This unexpectedly low value implies a weak inverse relationship between OUR and temperature, suggesting that temperature alone is not the dominant driver of interannual OUR variability. This corroborates results of previous studies (Jutras et al., 2020, 2023b) indicating that shifts in circulation dynamics (e.g., changes in LCW:NACW ratios) and variations of OM flux exert a more substantial influence on OUR than temperature. A more detailed comparison of the linear regression and $Q_{10}$ model fits is provided in the Supplemental Material (S6).

Other observational and modeling studies have suggested faster advection speeds than the one applied here (e.g., Gilbert, 2004; Rousseau et al., 2025). In our 1D framework, a doubling of the advection speed (e.g., to $u = 1 \times 10^{-2}$ m s⁻¹) would require a concomitant doubling of the OUR to maintain the same along-channel DO gradient. Discrepancies in OUR estimates from studies such as Benoit et al. (2006) and Lehmann et al. (2009) can largely be attributed to differences in methodology and assumptions. Both studies use an advection velocity of $1.0 \times 10^{-2}$ m s⁻¹, twice the value we employ in our analysis ($u = 5.0 \times 10^{-3}$ m s⁻¹). Since OUR in the LC deep layer is inversely related to the transit time of deep waters between the Cabot Strait and the head of the LC, adjusting their models to this lower velocity reduces their OUR estimates, bringing them closer to our value. For instance, the Benoit et al. (2006) high OM flux scenario yields an OUR of ~20 μmol kg⁻¹ yr⁻¹ when an $u = 5.0 \times 10^{-3}$ m s⁻¹ is applied (Bourgault et al., 2012; Jutras et al., 2020). They also assumed a vertical diffusivity coefficient ($K_Z$; see Table 1) that was two orders of magnitude greater than our tracer-based estimates (Stevens et al., 2024), a discrepancy that

enhances vertical oxygen supply and inflates their OUR estimate. Likewise, Lehmann et al. (2009)'s higher OUR can be recalculated under the same conditions (26.7 µmol kg$^{-1}$ yr$^{-1}$). Finally, the lower OUR reported by Savenkoff et al. (1996; ~30% less than our estimate) is likely due to differences in spatial resolution and methodology: their

estimate is based on the difference between only two, geographically separated, stations, whereas ours is derived from a model fit to data from multiple stations along the LC, providing greater spatial coverage and resolution. These comparisons illustrate the sensitivity of rate estimates on the choice of transport parameters. By informing our model with tracer-based observations, we reduce uncertainty in rate estimation and improve their comparability to both previous studies and future projections.

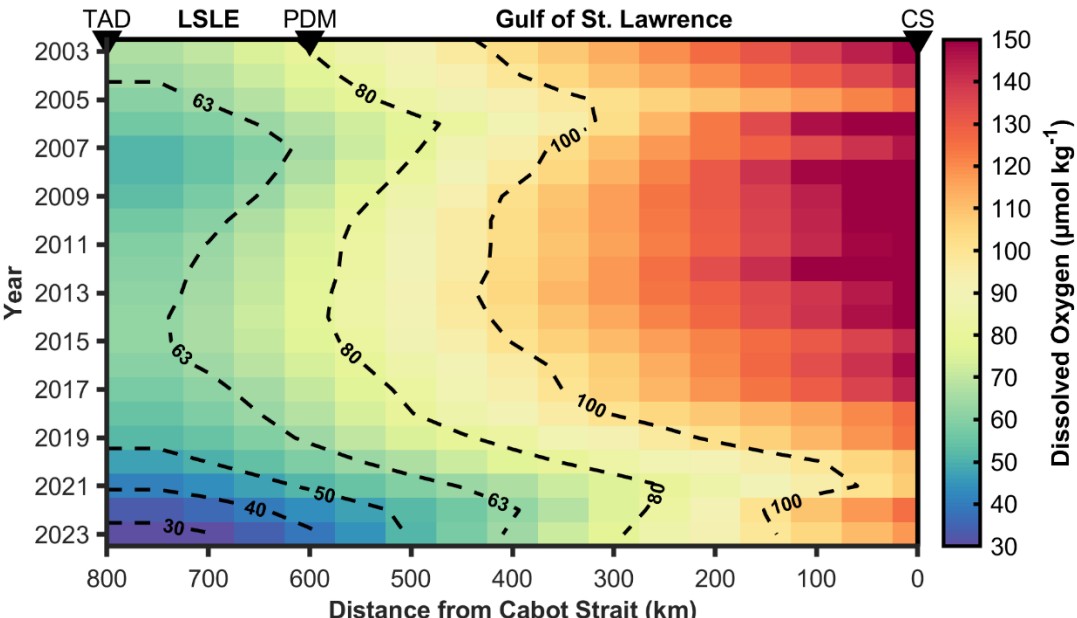


**Figure 5: Modeled deep-layer dissolved oxygen (DO) concentrations along the Laurentian Channel from 2003 to 2023 between the 27.15-27.30 kg m$^{-3}$ density layers. The x-axis shows the along-channel distance (in km) from the Cabot Strait (CS), the entry point of the deep-layer inflow to the Gulf of St. Lawrence from the western North Atlantic. Important landmarks along the transect are marked with black triangles: Cabot Strait (CS), Pointe-des-Monts (PDM), indicating**

**the transition between the Gulf of St. Lawrence (GSL) and the Lower St. Lawrence Estuary (LSLE), and Tadoussac (TAD), the head of the LC and LSLE. Dashed black contours highlight specific oxygen levels (µmol kg$^{-1}$) to aid identify spatial and temporal trends.**

### 3.4 Modeled DIC and δ$^{13}$C$_{DIC}$ results

Here, we describe the evolution of DIC and δ$^{13}$C$_{DIC}$ in the deep-layer core (27.15–27.3 kg m$^{-3}$) of the LC. For this

analysis, we exclusively use data from the TReX (2021–2023) dataset, as it possesses paired measurements of DIC and δ$^{13}$C$_{DIC}$. Initial boundary conditions at the Cabot Strait for both parameters are estimated from the intercept of an along-channel linear least-squares regression fit to observations from each sampling year (see Supplemental Material S4). The boundary DIC concentrations are determined to be 2230 ± 5 (2021), 2232 ± 6 (2022), and 2222 ± 11 (2023) µmol kg$^{-1}$, whereas δ$^{13}$C$_{DIC}$ values are 0.26 ± 0.07 (2021), 0.19 ± 0.08 (2022), and 0.09 ± 0.16 (2023) ‰.

Deep-layer DIC accumulation and dilution rates of $^{13}C_{DIC}$ for the 2021–2023 period are estimated from the previously-described least-squares fits (see description of the model in the Methodology section) and found to be 18.3 ± 2.5 µmol kg$^{-1}$ yr$^{-1}$ and -0.19 ± 0.02 ‰ yr$^{-1}$, respectively.

Given that $A_T$ varies conservatively with salinity ($A_T/S_p = 54 × S_p + 428$, R$^2$ = 0.90; see Supplemental Material S7) throughout the deep layer (>150m depth) of the LC, the contribution of benthic CaCO$_3$ dissolution to the deep-layer
DIC is assumed to be minor. This linear relationship suggests that $A_T$ variability is primarily driven by mixing, rather than CaCO$_3$ dissolution or precipitation or other $A_T$ generating processes (e.g., $A_T$ diffusion from the sediment in response to anaerobic respiration). CaCO$_3$ production in surface waters within the LSLE and GSL is limited to seasonal coccolithophore blooms in the Strait of Belle Isle (Fuentes-Yaco et al., 1997; Levasseur et al., 1994, 1997). Furthermore, with the exception of benthic foraminifera (Audet et al., 2023) and occasional small bivalves, the
inorganic carbon (IC) content of the LSLE sediments does not exceed 0.5% (dry weight) whereas the erosion of Ordovician-Silurian carbonates from the Mingan Archipelago and Anticosti Island increase the LC sediment IC content to ~3% (dry weight) in their immediate vicinity. These sediments are now overlain by oxygen-depleted, metabolically-acidified bottom waters (Mucci et al., 2011) and the accumulation of metabolic CO$_2$ generated by the aerobic degradation of OM in the surficial, oxic sediment generates CaCO$_3$-undersaturated porewaters and the
dissolution of bivalve shells and detrital carbonates (Nesbitt and Mucci, 2021). Whereas this benthic dissolution may contribute DIC and $A_T$ to the overlying waters, its influence on the DIC inventory of the LC is minimal throughout the GSL and LSLE. Furthermore, the dissolution of CaCO$_3$ minerals would enrich the $^{13}C_{DIC}$, partially offsetting the depletion caused by OM remineralization. The negative correlation between $δ^{13}C_{DIC}$ and DIC concentrations ($−6.6×10^{-3}$ ‰ per µmol of DIC) observed here supports the hypothesis that OM remineralization is the dominant
process controlling the DIC variability and $δ^{13}C_{DIC}$ signatures in the LC bottom waters.

From the intercept of the Keeling Plot relationship ($δ^{13}C_{DIC}/(1/DIC)$; Keeling, 1961), we estimate the average isotopic composition ($δ^{13}C_{DIC}$) of the remineralized OM to be -17.5 ± 1.5 ‰, typical of marine primary production. A similar value, -19.8 ‰, was reported for the particulate OM in the LSLE (Lucotte et al., 1991). The authors concluded that the majority of terrigenous OM transported into the Upper St. Lawrence Estuary by the St. Lawrence
River is transformed or recycled before reaching the LSLE and the LC. More recently, Lévesque et al. (2023) confirmed this hypothesis, showing that significant along-estuary transformation and retention of OM occurs, with 64-90% of particulate OM and 30-63% of dissolved OM being altered before entering the LSLE. Wang et al. (2025) report an average $δ^{13}C_{org}$ of -22.4 ± 1.4 ‰ for the particulate OM in the GSL. In other words, only a fraction of the terrestrial OM delivered by the main freshwater tributaries (i.e. St. Lawrence and Saguenay Rivers) reaches the LC,
and most of the OM respired at depth in the LC originates from marine primary productivity. However, it has been recently shown that microbial degradation can shift the isotopic composition of terrestrial OM to mimic a marine-like $δ^{13}C$ signature (Goranov et al., 2025). As such, while our reported $δ^{13}C_{DIC}$ data supports a predominantly marine OM source within the LC, the possibility of altered terrestrial inputs cannot be fully excluded.

We use the modelled OUR and DIC accumulation rates to estimate an average respiration quotient ($r_{-O_2:C}$ = -
OUR/DIC accumulation rate) of 1.15 ± 0.21. This is slightly lower than the classical Redfield ratio (138:106 = 1.30;

Redfield et al., 1963), but stoichiometric deviations are not uncommon, particularly in estuarine or coastal environments that are impacted by continental and anthropogenic OM inputs. For example, Körtzinger et al. (2001) reported a r-$_{O_2:C}$ of ~1.34 in the North Atlantic, whereas Moreno et al. (2020, 2022) observed a global oceanic range between 0.99 and 1.19, with an average of 1.16, more closely matching our findings. Our estimate is consistent with results of both studies given the sizable, propagated uncertainty of r-$_{O_2:C}$ (± 0.21). The uncertainty likely reflects both the natural variability of the stoichiometry of the respired OM and limitations of our model (i.e. rates being an along-channel average). Anderson and Sarmiento (1994) emphasized that, whereas the Redfield ratio provides a practical framework, remineralization stoichiometry can vary regionally. This may be particularly relevant to the LC, as Pascal et al. (2025) reported that fixed-nitrogen loss occurs at DO concentrations <57 µmol kg$^{-1}$, a threshold commonly reached in the LSLE. This reduces the oxygen demand associated with remineralization and would contribute to a lower r-$_{O_2:C}$. Variations in the composition of OM could further influence respiration stoichiometry. Tanioka and Matsumoto (2020) demonstrated that r-$_{O_2:C}$ depends on the macromolecular composition of OM, ranging from ~1.0 for carbohydrate-rich substrates to ~1.4 for lipid-rich substrates. Proteins and carbohydrates, that have lower C:N ratios, are preferentially remineralized at shallower depths, leaving lipid-rich OM with higher C:N ratios to be exported and respired at depth (Hedges et al., 2002). These interpretations suggest that the observed r-$_{O_2:C}$ within the LC reflects a combination of stoichiometric variability, the influence of denitrification or other anaerobic processes, and depth-dependent changes in OM composition.

**3.5 Implications for possible mitigation**

Through multiple simulations using a constant, arbitrary inflow concentration and a fixed OUR, it was determined that approximately 10 years are required for the dissolved metabolites (DO, DIC) in the deep layer of the LC to reach a new steady state following a perturbation in the inflow composition. The simulations were conducted with step-wise increases in run time (1 year, 2 years, 3 years, and so on), allowing the system to evolve progressively. A stable, asymptotic interannual concentration was observed at $x$ = 800 km after 10 years of simulation, corresponding to approximately two transit times.

Following the declining contribution of LCW to the deep-layer inflow into the GSL after 2020 and the historically low minimum DO concentrations measured at the head of the LC (Jutras et al., 2023b), mitigation has become a topic of interest. Wallace et al. (2023) suggested that pure oxygen, a by-product from three proposed green hydrogen/ammonia production plants in Stephenville (Newfoundland), could be injected into the inflowing deep layer just inside the Cabot Strait to "re-oxygenate" the LC. They estimated that these plants would produce up to 5 × 10$^5$ tonnes (1.56 × 10$^{10}$ moles) of pure oxygen per year as an operational by-product. Here, we make an initial assessment of this reoxygenation scenario by modifying the boundary condition in our advection-diffusion model. Instead of injecting oxygen at a discrete grid point, we assumed a continuous distribution along the deep-layer inflow. Specifically, the additional oxygen input was applied over a representative cross-section of the LC at Cabot Strait (30 km width × 100 m depth × 1 m thickness). The injected oxygen mass (in tonnes yr$^{-1}$) was converted to a concentration increase (in µmol kg$^{-1}$) using a seawater density of 1027.25 kg m$^{-3}$, yielding an estimated increase of

32 μmol kg$^{-1}$ at the boundary. Two scenarios were considered for the period from 2024 onwards, using the 2023 transect to set initial conditions. Scenario (a) represents a baseline case where no mitigation is implemented, and inflow [DO] remains fixed at ~115 μmol kg$^{-1}$ (Fig. 2). Scenario (b) explores the impact of injecting 5 × 10$^5$ tonnes per year of pure oxygen into the inflow, raising the boundary [DO] to ~147 μmol kg$^{-1}$, comparable to levels observed prior to 2018 (Fig. 2). Whereas this approach provides an initial assessment of feasibility, it does not account for potential oxygen losses through alternative flow pathways (e.g. the Esquiman Channel or through the Cabot Strait) or localized vertical mixing effects.

For scenario (a), at steady-state, the hypoxic zone extends approximately 400 km seaward from the head of the LC near Tadoussac to the vicinity of Anticosti Island (Fig. 6a). DO concentrations drop to as low as ~25 μmol kg$^{-1}$ near Tadoussac, reflecting a permanent state of severe hypoxia in the LSLE. Since 2020, the deep-layer inflow has been composed exclusively of NACW (Jutras et al., 2020, 2023b), with this new mixture progressively propagating upstream towards the LSLE. The hypoxic zone had only reached the western boundary of the GSL by 2021 (Jutras et al., 2023b), indicating a delayed response in oxygen depletion relative to the inflow shift. If the LCW contribution to the Cabot Strait inflow remains negligible, our projections suggest that the magnitude and areal extent of the hypoxic zone will continue to expand until ~2030, approximately 10 years after the shift to an NACW-dominated inflow. Beyond this time, if all other factors remain constant (e.g., OM and nutrient inputs from the tributaries, NACW DO concentrations), the hypoxic zone is not expected to evolve further, as a new steady state will have been established in response to the prevailing inflow concentrations.

In scenario (b), the hypoxic front progressively migrates westward toward the head of the LC within the first 5 years. By year 10 (2033), when the system has reached a new steady state, the hypoxic zone would have migrated approximately 250 km from its 2023 condition (near Anticosti Island), retreating into the LSLE near Pointe-des-Monts. These conditions are similar to those observed within the deep layer at the turn of the millennium (see Fig. 2 in Jutras et al., 2023b).

Although the results of scenario (b) are promising, they do not provide a prognosis of a complete recovery. Additional inputs of oxygen would be required to bring and maintain the bottom-water [DO] of the entire LSLE above the hypoxic threshold (62.5 μmol kg$^{-1}$). We estimate the amount of oxygen required to accomplish this with our simplified (linear fit) advection-diffusion equation with a source term:

$$C(x) = C(0) + \frac{S \times x}{u} \tag{5}$$

that can be rearranged as:

$$C(0) = C(x) - \frac{S \times x}{u} \tag{6}$$

where $C(x)$ is the desired DO concentration at the end of the channel ($x$ = 800 km). Hence, $C(800)$ is set at 63 μmol kg$^{-1}$, $S$ is our sink term (mean OUR) of -21.1 μmol kg$^{-1}$ yr$^{-1}$ converted to -6.7 × 10$^{-7}$ μmol kg$^{-1}$ s$^{-1}$, x is the distance to the end of the channel (800 km), and u is the along-channel advection velocity (5 × 10$^{-3}$ m s$^{-1}$). Under this scenario,

the boundary condition, or $C(0)$, would have to be set to and maintained at ~170 μmol kg$^{-1}$ which is +55 μmol kg$^{-1}$ relative to the 2023 boundary concentration. Such an inflow [DO] would require the injection of ~8.3 × 10$^5$ tonnes yr$^{-1}$ of oxygen, approximately 70% more than the production capacity of the proposed Stephenville plants. To return to 1930 bottom-water DO levels (~125 μmol kg$^{-1}$; Gilbert et al., 2005) at the head of the LC, the boundary DO concentration would have to exceed 230 μmol kg$^{-1}$ (+115 μmol kg$^{-1}$) effectively doubling the current inflow concentration. This would require an injection of ~1.8 × 10$^6$ tonnes yr$^{-1}$ of oxygen.

Whereas the large-scale injection of by-product oxygen is a promising strategy to remediate bottom-water hypoxia in the LC, the proposed production at Stephenville only allows for restoration to year 2000 conditions (or at the beginning of the model runs). For the system to recover further, direct oxygen injection would have to be drastically increased in magnitude, or be combined with other measures, including the implementation of government regulations limiting anthropogenic inputs of OM and nutrients that drive autochthonous OM production in the estuary. The potential impact of increasing bottom-water temperatures on microbial respiration rates must also be considered but, as discussed earlier, temperature variations account for only ~17% of the interannual variability in OUR ($R^2$ = 0.171), based on our linear regression analysis, whereas a $Q_{10}$-based approximation yields a best-fit $Q_{10}$ value of 0.33, suggesting that temperature alone exerts a limited influence on the OUR variability. Since the deep-layer inflow is now entirely comprised of NACW, future temperature increases are expected to slow down relative to previous decades (Galbraith et al., 2024) but may continue to scale (at a slower rate) as a consequence of global warming. Surface ocean temperatures, including those of the NACW during formation, will continue to increase as the oceans absorb more than 90% of the heat trapped by greenhouse gases (Intergovernmental Panel on Climate Change, 2021). These factors underscore the need for integrating different and additional management strategies to address both external/regional drivers of deoxygenation as well as local inputs of OM and nutrients.

Whereas these estimates provide valuable insights, it is important to acknowledge the inherent assumptions and simplifications in the model. These include steady-state conditions, spatial homogeneity within the deep layer, and uniform biogeochemical rates, all of which introduce some degree of uncertainty in the model results. We also assume that all supplied oxygen is dissolved and retained within the deep layer inflow (i.e. 100% efficiency). This, however, may ultimately depend on location and timing of injection. Additionally, our approach does not account for vertical diffusivity, which, over decadal time scales, would influence the DO concentration throughout the LC but especially at the head of the channel where complex tidal phenomena, including internal tides and strong flows over the steep sill, contribute to significant mixing between near-surface waters and deeper saline layers (Bluteau et al., 2021; Cyr et al., 2015; Ingram, 1983). Moreover, the model simplifies a time and space varying horizontal circulation that could be influenced by features such as seasonal shifts in transportation pathways and mixing intensity (Rousseau et al., 2025). In contrast to previous fjord reoxygenation efforts that pumped surface water to depth to reduce vertical stratification and promote deepwater flushing (Stigebrandt et al., 2015), proposed interventions for the LC rely on direct injection of pure oxygen into the deep layer inflow (Wallace et al., 2023) using density-compatible aeration systems (e.g., bubble plume diffusers, turbine injectors, or Speece Cone-style

contact chambers). When combined with such injection systems and given the much deeper pycnocline of the LC (~150–200 m), we assume, in our first approximation, that vertical stratification would be preserved. Nonetheless, we recommend that future studies address oxygen plume dynamics to better assess injection efficiency and oxygen loss, and that more complete 3D numerical models be developed to refine projections and evaluate mitigation
strategies and their broader impacts.

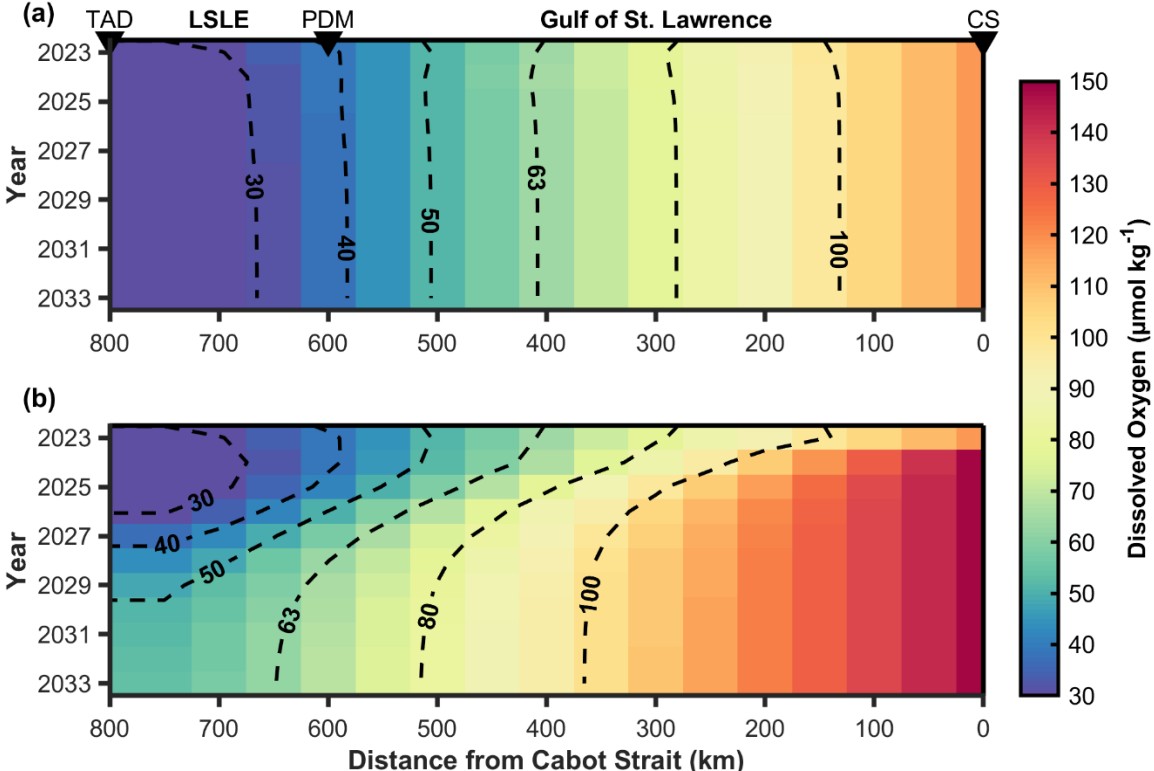

**Figure 6: Projected dissolved oxygen (DO) concentrations along the Laurentian Channel (LC) from 2023 to 2033 under two scenarios. (a) No mitigation, showing natural trends in DO levels without intervention. (b) An oxygen injection**
**scenario with 5 × 10⁵ tonnes per year, illustrating the potential improvement in DO concentrations over time. The x-axis represents the distance (in km) from the Cabot Strait, marking the deep-layer inflow point. Dashed black contours indicate key DO concentration levels (μmol kg⁻¹) to highlight spatial and temporal trends.**

## 4 Conclusions

In this study, we present average, along-channel, DO utilization, DIC accumulation, and $^{13}C_{DIC}$ dilution rates from
the fitting of a tracer-informed 1D Advection-Diffusion model (Stevens et al., 2024) to field measurements from four cruises during the 2021-2023 TReX project. Based on two decades of historical DO and physicochemical data, we demonstrate that the OUR in the deep layer of the LC was relatively constant between 2003 and 2023 at 21.1 ± 2.5 μmol kg⁻¹ yr⁻¹, consistent with some previously published estimates (Bourgault et al., 2012; Jutras et al., 2020), but incompatible with others that used different advection and mixing parameters (Benoit et al., 2006; Lehmann et
al., 2009).

Deep-layer DIC accumulation and $^{13}C_{DIC}$ dilution rates were estimated at $18.3 \pm 2.5$ μmol kg$^{-1}$ yr$^{-1}$ and $-0.19 \pm 0.02$ ‰ yr$^{-1}$, respectively. The relationship between $^{13}C_{DIC}$ dilution and DIC accumulation yields an estimate of the impact of in-channel remineralization on the $\delta^{13}C_{DIC}$ of $-6.6 \times 10^{-3}$ ‰ per μmol of metabolic DIC accumulated. Hence, we derive a respiration quotient (r-$_{O_2:C}$ = -OUR/DIC accumulation rate) of $1.15 \pm 0.21$, a value that is slightly lower than the classical Redfield ratio (1.30). This discrepancy, although within the uncertainty, may reflect the compositional variability of the settling OM, the potential contribution of water column denitrification, and depth-dependent alteration of OM composition.

We determine that the time required for the system to reach a new steady state, following a change in the DO concentration of the deep waters that flow into the GSL from the Atlantic Ocean through Cabot Strait, is approximately 10 years (~2 transit times). This study utilizes the 1D model as a predictive tool to explore the mitigation scenario presented by Wallace et al. (2023), simulating an injection of $5 \times 10^5$ tonnes of oxygen per year into the boundary layer at Cabot Strait. Whereas this injection could push the hypoxic zone ($>62.5$ μmol kg$^{-1}$) landward by ~250 km to Pointe-des-Monts, a considerably larger oxygen input ($8.3 \times 10^5$ tonnes yr$^{-1}$) would be required to eliminate bottom-water hypoxia outwardly from the LSLE. These findings emphasize the importance of better constraining physical advection and mixing parameters to predict the distribution of metabolites and changes within the GSL. The observational dataset supporting these findings spans two decades and provides a unique baseline for assessing long-term variability in the cycling of DO and DIC. The simple model used here highlights the value of tracer-validated models but also emphasises the need for development of more sophisticated modeling efforts to adequately evaluate mitigation strategies. It should be noted that the rates (OUR, DIC accumulation, and $^{13}C_{DIC}$ dilution) presented in this study are average fits to the deep-layer inflow core (27.15-27.3 kg m$^{-3}$ isopycnals) in the LC, assuming a constant advection speed and diffusivity, over an 800 km transect. Local and seasonal variability in hydrodynamics or remineralization rates, the relative importance of SOD in relation to pelagic OUR, and the importance of $K_Z$ cannot be considered in our simple model and should be the focus of future efforts.

**Data availability**

The observational dataset used in this study includes measurements collected during the TReX project (2021-2023) and historical data (2003-2020), and forms part of a larger, integrated data product documenting a 20-year biogeochemical time series in the St. Lawrence Estuary, Gulf, and Saguenay Fjord. This data product is archived through the Canadian Integrated Ocean Observing System – St. Lawrence Global Observatory (CIOOS-SLGO) under the DOI https://doi.org/10.26071/d6f3fdfc-788d-48ff (Nesbitt et al., 2025). Whereas the metadata are publicly accessible, the data are currently under moratorium and will be released following the submission of a companion data manuscript to be submitted to *Earth System Science Data*. In the meantime, the subset of data supporting this study is available from the corresponding author upon reasonable request.

**Author contributions**

WAN and DWRW conceived the study. WAN, SWS, LG, TT, GC, and DWRW coordinated and conducted at-sea campaigns. SWS set up the 1D Adv-Diff model. WAN and SWS conducted the analyses. AOM provided a significant and invaluable time series of biogeochemical data. WAN wrote the manuscript with writing and editorial contributions from all authors.

**Competing interests**

The authors declare that there are no conflicts of interest.

**Acknowledgements**

We would like to extend our thanks to Reformar, the Captain and crew of the R/V Coriolis II, and to the Department of Fisheries and Oceans for providing access to ship-time and cruise data. We would specifically like to thank Dr. Marjolaine Blais for allowing us to participate on the 2022 Atlantic Zone Monitoring Program (AZMP) cruise and for sharing the supporting hydrographic data, Dr. Ludovic Pascal for inviting us on the 2023 PLAINE campaign, and Dr. Mathilde Jutras for providing estimates of inflowing boundary conditions prior to 2019. We would like to thank Dr. Anders Stigebrandt, Dr. Xianghui Guo, and an anonymous reviewer for their constructive and helpful comments during the open discussion period. Finally, we would like to extend special thanks to the technical participants in the TReX project: Caroline Fradette, Adriana Reitano, Chukwuka Orji, Sara Wong, and Olivier Turcotte. Execution of this project would not have been possible without them.

**Financial support**

Financial support for the TReX project was provided by the Marine Environmental Observation, Prediction and Response (MEOPAR) network of centers of excellence as well as the Réseau Québec Maritime (RQM) and its Odyssée Saint-Laurent ship-time project. Partial support for student personnel and technical assistance was provided by a NSERC Discovery Grant to DWRW. We would like to recognize the financial support of the Government of Nova Scotia to WAN in the form of a four-year NSGS-D scholarship. SWS was supported by a TReX Graduate Award, a UBC Four-Year fellowship, and a postdoctoral fellowship from the Tula Foundation.

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
