# Peer review of "The coupled oxygen and carbon dynamics in the subsurface waters of the Gulf and Lower St. Lawrence Estuary and implications for artificial oxygenation"

_EGUsphere, 2025_

## Author Comment (AC1)

Dear Dr. Stigebrandt,

We appreciate your thoughtful and constructive review of our manuscript. Your comments helped us improve the clarity and depth of this study.

**Comment 1**: The rate of change of DO along the flow path depends on (i) the rate of oxygen consumption OUR by mineralization of OM, (ii) the rate of oxygen supply by inflow through Cabot Strait, which is determined by the advective flow speed u and the concentration of DO of the inflowing water (boundary concentration), and (iii) the rate of supply of oxygen by turbulent vertical diffusion from the oxygen rich Cold Intermediate Layer overlying the deepwater. Turbulent vertical diffusion is known to take place essentially at the bottom boundary, where most of the vertical mixing occurs due to breaking internal waves, often driven by internal tides generated at sloping bottoms and steps in the bottom. Vertical mixing at bottom boundaries creates horizontal buoyancy gradients that drive transversal circulation, not described by a 1D model, that distributes the effects of mixing to the whole water body. Changes of DO due to changes in the rate of change of turbulent vertical mixing are hard to show. If the turbulent mixing is driven by the internal tide, it may change if the vertical stratification changes. This could be discussed in the manuscript.

This is an important point, and we recognize the significance of internal tide-driven mixing in the bottom boundary layer and its potential influence on vertical and horizontal oxygen fluxes into the deep layer. Previous studies (Cyr et al., 2011, 2015) have shown that while interior mixing accounts for the majority (70%) of CIL erosion near Rimouski (proximal to the head of the Laurentian Channel in the Lower Estuary), significant boundary-layer turbulence also occurs where the CIL intersects with the sloping seafloor. It was also noted that the relative contribution of these processes is likely to shift even further towards interior mixing dominance in the open Gulf. Whereas our 1D model neglects vertical diffusivity, this simplification is supported by tracer-based observations from Stevens et al. (2024) in which the authors report basin-wide effective vertical diffusivities on the order of $10^{-5}$ $m^2$ $s^{-1}$ (6.5 x $10^{-6}$ $m^2$ $s^{-1}$ in the interior, 1.5 x $10^{-5}$ $m^2$ $s^{-1}$ in boundary regions). Although localized enhancements in vertical mixing were observed near the basin slopes, these effects were spatially limited. These findings are consistent with earlier microstructure measurements (Cyr et al., 2011) and support our use of a steady-state 1D advection-diffusion framework focused on along-channel dynamics. That said, such localized mixing may contribute to spatial heterogeneity in oxygen concentrations and secondary circulation, particularly near topographic features, and we now acknowledge this as a potential limitation of our approach. A brief discussion of these points and their significance now appears in Section 3.2 of the revised manuscript.

**Comment 2:** The 1D model is tuned using historical data, and data from the large-scale tracer experiment TReX in the Bay of St Lawrence. It is required that the model can describe the distribution of DO along its path. If it can, one may have confidence in model results when changing the boundary concentration of DO at Cabot Strait.

We agree that confidence in the model's projections depends on its ability to reproduce present-day DO distributions. The 1D advection-diffusion model utilized in this study is constrained using physical parameters derived from the TReX tracer experiment (Stevens et al., 2024), specifically the along-channel advection velocity and horizontal diffusivity. For each run (or year), the model is fit to observed, along-channel DO concentrations using a least-squares fit repeated over 1000 iterations, providing a robust fit to the data. The resulting model output captures the observed along-channel DO gradient in both magnitude (amount of DO consumed within the channel) and slope (OUR). This supports the validity of investigating the impact of possible mitigation scenarios upon varying boundary conditions.

**Comment 3:** Horizontal diffusion, but not vertical diffusion, is included in the model. The argument for discarding vertical diffusion is that is has a time scale of 30 years while the horizontal time scale is 5 years. However, the vertical time scale is estimated using the vertical diffusivity $K_z = 1 \times 10^{-5}$ $m^2 s^{-1}$. The horizontal mean $K_z$ is maybe larger because one may expect high values at the boundaries (hot mixing spots). With $K_z = 1 \times 10^{-4}$ $m^2 s^{-1}$, the vertical time scale would be only 3 years. This should be discussed in the manuscript because vertical diffusion possibly may provide a significant contribution to the DO budget of the deepwater.

We agree that the role of vertical diffusion warrants a more in-depth discussion. This was partially addressed in our response to "Comment 1" above. In the manuscript, we estimate a vertical diffusive timescale of ~30 years based on a representative vertical diffusivity $K_Z = 10^{-5}$ m$^2$ s$^{-1}$, consistent with basin-wide estimates by Stevens et al. (2024) based on results of the TReX tracer experiment. Stevens et al. (2024) also report effective vertical diffusivities of ~$6.5\times10^{-6}$ m$^2$ s$^{-1}$ in the interior and up to $1.5\times10^{-5}$ m$^2$ s$^{-1}$ near the boundaries of the Laurentian Channel, values that are consistent with previous microstructure measurements in the Lower Estuary (Cyr et al., 2011). Whereas localized "hotspots" of mixing may exhibit even higher $K_Z$ values (e.g., $10^{-4}$ m$^2$ s$^{-1}$), these regions are likely spatially constrained. Thus, basin-wide averages seem appropriate to assess the large-scale DO budget in the Laurentian Channel. Whereas we acknowledge that vertical diffusion could affect DO distributions, especially in regions with rough topography, our decision to neglect vertical diffusivity in the 1D framework is justified by the dominance of along-channel transport over large spatial scales. As noted in our response to "Comment 1", a brief discussion of these issues was added to Section 3.2 of the revised manuscript.

**Comment 4:** The model describes quite well the observed year to year changes in DO in the deepwater using only known changes of the DO concentration in the Cabot Strait. The model uses a constant UOR. One would expect that UOR might be greater in the inner part of the St. Lawrence River Estuary due to possibly greater production of OM here due to nutrient supply by the river. It would be interesting if the authors could discuss the sensitivity of model results to the assumption of a constant UOR. It would also be interesting to know if there are large variations in the annual supply of nutrients from the St. Lawrence River, and the expected annual supply of OM to the deepwater.

We agree that the assumption of a spatially uniform OUR may overlook heterogeneity in primary production and organic matter supply across the St. Lawrence Estuary and Gulf. Below, we address the model's sensitivity to this assumption and summarize relevant literature on the variability of nutrient and organic matter supply. To evaluate the impact of a spatially heterogeneous OUR, we modified the 1D advection-diffusion model and doubled the remineralization rate (S) over the final 200 km of the Laurentian Channel (approximately corresponding to the LSLE), while retaining the best-fit S value elsewhere. This scenario was intended to represent enhanced respiration in nearshore regions due to elevated OM supply. The simulation was run to steady state (over a 10-year period or roughly 2 transit times) and used 2022 as a test year due to its extensive spatial data coverage. As shown in the figure below (Fig. R1), introducing a spatially variable S leads to slightly lower predicted DO concentrations in the inner estuary, particularly beyond 600 km. While the reduction in least-squares misfit is modest (–2.3%), the fit appears visibly improved in the LSLE, suggesting a slightly better spatial alignment between model and observations. This suggests that, although spatial heterogeneity in remineralization rates likely exists, the large-scale DO trends are effectively captured by a spatially averaged OUR in the present model configuration that integrates over regional scales. With respect to the variability of nutrient and OM supply, several studies point to distinct seasonal and spatial patterns in source contributions and productivity across the system. For instance, Savenkoff et al. (2001) and Jutras et al. (2020) show that most of the nitrate input to the LSLE during the summer originates from deep water upwelling at the head of the Laurentian Trough. In contrast, Bluteau et al. (2021) report that during the winter, the primary nitrate source is fluvial, owing to reduced upstream production. The origin of the settling organic matter also appears to vary along the estuarine gradient. Benoit et al. (2006) and Wang et al. (2025) show that sediments (and, presumably deep waters) of the Upper Estuary receive more allochthonous organic matter, which tends to be less labile, whereas autochthonous organic matter fluxes dominate in the LSLE and this OM is more readily respired. This is consistent with the inferred (from conversion of multi-mission remote sensing datasets to daily Chl a concentrations between 1998-2019) spatial and seasonal distribution of Chl a (in the Estuary and Gulf), including the presence of a mid-estuary maxima (Laliberté and Larouche, 2023), and would suggest a spatially varying remineralization potential. Nonetheless, our model, which resolves large-scale advection and integrates over relatively long spatial and temporal scales, agrees well with observed deepwater DO concentrations using a single OUR value. This suggests that, whereas OUR heterogeneity exists, its impact on the deepwater DO distribution is reduced by the large spatial and temporal scales of the channel, which tend to smooth out local heterogeneity. Nonetheless, we agree that spatially resolved models would be required to further investigate DO dynamics. A brief discussion of primary productivity heterogeneity and the OUR sensitivity analysis was added to Section 3.3 of the revised manuscript. Figure R1 was added to Supplemental Material S8 and referred to in the revised manuscript.

[Figure]

**Figure R1: Sensitivity of the 1D advection-diffusion model to spatial variability in the oxygen utilization rate (OUR). The model is initialized with observed deepwater O₂ concentrations at Cabot Strait and run forward with either a spatially uniform OUR (solid blue line; fit to observations from 2022 (black dots)) or a doubled OUR in the inner 200 km of the Laurentian Channel (dashed red line), approximating elevated remineralization in the LSLE where the supply of allochthonous OM is enhanced. Regional labels highlight key longitudinal transitions along the Laurentian Channel: TAD = Tadoussac, LSLE = Lower St. Lawrence Estuary, PDM = Pointe-des-Monts, and CS = Cabot Strait. The spatial pattern of [DO] is well captured in both cases, with the doubled-OUR scenario yielding only a ~2% decrease in misfit.**

**Comment 5:** The physics of the model, i.e. the current speed u and the horizontal diffusivity $K_H$, has been calibrated using results from the transient plume of the tracer experiment TReX. Since the duration of the tracer experiment was shorter than the residence time of the deepwater it was necessary to include horizontal diffusivity to describe the observed spreading of the tracer. For a quasi-steady description of DO it might possibly be more important to include vertical diffusion since vertical diffusion might contribute to the DO budget of the deepwater. The authors should discuss this and estimate the uncertainty of model results due to the explicit ignorance of vertical diffusion.

The vertical diffusion may contribute to the deep layer oxygen budget. Here, we perform a rough estimation of its magnitude using Fick's Law:

$$F = -K_Z \times \frac{\partial C}{\partial z}$$

We define the concentration gradient ($\partial C$) as the Deep Layer [DO] (with respiration removed, i.e. the boundary condition) – the average CIL [DO]. The depth of the diffusive layer ($\partial z$) was set to 100 m (the same depth used for the dimensional analysis). We used data from 2022 (as this was a robust year for data collection) for an example calculation and explored the average estimated flux based on 3 scenarios: 1) The basin-wide effective $K_Z$ of $10^{-5}$ (Stevens et al., 2024), 2) The boundary mixing effective $K_Z$ of $1.5 \times 10^{-5}$ (Cyr et al., 2011; Stevens et al., 2024), and

finally 3) The hypothetical hotspot $K_Z$ of $10^{-4}$. The $\partial C$ for 2022 was estimated to be ~129 μmol kg$^{-1}$ (CIL aver DO = 254 μmol kg$^{-1}$, and DL inflow (respiration removed) = 125 μmol kg$^{-1}$). The fluxes for the 3 scenarios are as follows:

1) Basin-wide average: ~1.1 μmol kg$^{-1}$ yr$^{-1}$ (5.3% of OUR).
2) Boundary-enhanced mixing: ~1.7 μmol kg$^{-1}$ yr$^{-1}$ (7.9% of OUR).
3) Hot spot scenario: ~11.1 μmol kg$^{-1}$ yr$^{-1}$ (53% of OUR).

These estimates show that even under enhanced mixing conditions at the boundaries, the contribution of vertical diffusion remains modest compared to the horizontal advection and remineralization dynamics resolved in the 1D model. This supports our decision to omit vertical diffusion in the core framework, while acknowledging that it may introduce spatial heterogeneity in DO near topographic boundaries / slopes and more significantly in localized hot spots. We added a paragraph to the expanded discussion on $K_Z$ in Section 3.2 of the revised manuscript.

**References**

Benoit, P., Gratton, Y., and Mucci, A.: Modeling of dissolved oxygen levels in the bottom waters of the Lower St. Lawrence Estuary: Coupling of benthic and pelagic processes, Marine Chemistry, 102, 13–32, https://doi.org/10.1016/j.marchem.2005.09.015, 2006.

Bluteau, C. E., Galbraith, P. S., Bourgault, D., Villeneuve, V., and Tremblay, J.-É.: Winter observations alter the seasonal perspectives of the nutrient transport pathways into the lower St. Lawrence Estuary, Ocean Science, 17, 1509–1525, https://doi.org/10.5194/os-17-1509-2021, 2021.

Cyr, F., Bourgault, D., and Galbraith, P. S.: Interior versus boundary mixing of a cold intermediate layer, Journal of Geophysical Research: Oceans, 116, https://doi.org/10.1029/2011JC007359, 2011.

Cyr, F., Bourgault, D., and Galbraith, P. S.: Behavior and mixing of a cold intermediate layer near a sloping boundary, Ocean Dynamics, 65, 357–374, https://doi.org/10.1007/s10236-014-0799-1, 2015.

Jutras, M., Mucci, A., Sundby, B., Gratton, Y., and Katsev, S.: Nutrient cycling in the Lower St. Lawrence Estuary: Response to environmental perturbations, Estuarine, Coastal and Shelf Science, 239, 106715, https://doi.org/10.1016/j.ecss.2020.106715, 2020.

Laliberté, J. and Larouche, P.: Chlorophyll-a concentration climatology, phenology, and trends in the optically complex waters of the St. Lawrence Estuary and Gulf, Journal of Marine Systems, 238, 103830, https://doi.org/10.1016/j.jmarsys.2022.103830, 2023.

Savenkoff, C., Vézina, A. F., Smith, P. C., and Han, G.: Summer Transports of Nutrients in the Gulf of St. Lawrence Estimated by Inverse Modelling, Estuarine, Coastal and Shelf Science, 52, 565–587, https://doi.org/10.1006/ecss.2001.0774, 2001.

Stevens, S. W., Pawlowicz, R., Tanhua, T., Gerke, L., Nesbitt, W. A., Drozdowski, A., Chassé, J., and Wallace, D. W. R.: Deep inflow transport and dispersion in the Gulf of St. Lawrence revealed by a tracer release experiment, Commun Earth Environ, 5, 1–13, https://doi.org/10.1038/s43247-024-01505-5, 2024.

Wang, Y., Ahad, J. M. E., Mucci, A. O., Gélinas, Y., and Douglas, P. M. J.: Large burial flux of modern organic carbon in the St. Lawrence estuarine system indicates a substantial atmospheric carbon sink, Earth and Planetary Science Letters, 652, 119204, https://doi.org/10.1016/j.epsl.2025.119204, 2025.

---

## Author Comment (AC2)

Dear Dr. Guo,

We would like to extend our thanks for your constructive review of our manuscript. Below we respond point by point to each of your comments.

**Comment 1:** The simulation is based on steady state. However, when the pure oxygen is injecting, the stratification is destroyed. This may lead to the sequence that the model parameters may no longer be appropriate. Additionally, is there oxygen leakage during the injection?

In previous re-oxygenation programs, such as the By Fjord in Sweden (Stigebrandt et al., 2015), oxygen-rich surface water was pumped into the basin over 2.5 years. Although this did not cause destratification, despite a near-surface halocline at ~15 m depth (although there is a substantial density gradient ($\Delta\rho$ <8 kg m$^{-3}$)) across a relatively shallow water column (<50m), the injection did reduce the density of the deep water. The weakened stratification enhanced the frequency of deep-water renewal from neighbouring basins, acting as an important mechanism for sustaining re-oxygenation in the fjord. Similarly, feasibility assessments for re-oxygenation in Hood Canal, Washington (Beutel and Wilson, 2005), emphasized that maintaining vertical stratification is possible using specialized delivery systems such as Speece Cones (submerged contact chambers) or bubble plume systems configured to minimize vertical mixing. These systems have been shown to deliver oxygen at depth with minimal disturbance to the stratification. In contrast, the site of our study, the Laurentian Channel, exhibits a more modest density gradient ($\Delta\rho$ ~0.6 kg m$^{-3}$) between the Cold Intermediate Layer and the Deep Layer inflow, but its pycnocline lies much deeper (~150-200 m), with bottom depths of 275–500 m. The greater depth and isolation from surface forcing (e.g., wind and wave energy) increases resistance to vertical mixing. Proposals to reoxygenate the Laurentian Channel are based on the direct injection of pure oxygen into the deep layer via an appropriate aeration system, such as those mentioned above (Wallace et al., 2023). Likewise, our model assumes that oxygen is introduced in fully dissolved form, well below the pycnocline, via a density-compatible delivery system such as turbine injection, bubble plume diffusers, or Speece Cone-style contact chambers. Whereas our model does not resolve short-term mixing processes at the injection site, it provides a first approximation to the downstream steady-state impact of oxygenated inflow under the assumption of constant transport parameters. We acknowledge that, in a real-world situation, some loss (oxygen leakage) could occur due to bubble rise or outgassing, but our model assumes 100% retention of the added oxygen in the deep layer, a potential limitation. We added a few sentences (Section 3.5) to the revised manuscript recommending that future work incorporate plume-resolving dynamics to evaluate injection efficiency and potential leakage pathways. A brief paragraph was added to Section 3.5 of the revised manuscript.

**Comment 2:** Lines 47–49: The number of '(' and ')' are different…

We corrected this punctuation error in the revised manuscript.

**Comment 3:** Lines 358–359: From the DIC data of 2021, 2022 and 2023, I don't agree that the DIC accumulation rate is 18.3 ± 2.5 μmol kg$^{-1}$ yr$^{-1}$.

We would like to clarify that the DIC accumulation rate reported represents the average along-channel accumulation rate as the deep water transits from Cabot Strait to the head of the Laurentian Channel. The value is derived from model-fits to observed DIC distributions using a least-squares approach within the framework of a 1D advection-diffusion model, that was applied individually to the years 2021, 2022, and 2023. These fits yield annual accumulation rates of 19.9, 14.8, and 20.1 μmol kg$^{-1}$, respectively. The reported value of 18.3 μmol kg$^{-1}$ yr$^{-1}$ represents the mean of these three rates with ± 2.5 μmol kg$^{-1}$ yr$^{-1}$, reflecting the interannual standard deviation. To further characterize the accuracy of this estimate, we now report the 90% confidence interval ([14.1, 22.4] μmol kg$^{-1}$ yr$^{-1}$) on the mean, calculated using a t-distribution. The 90% confidence interval was selected to provide a practical estimate of uncertainty, given the small number of modeled years (n = 3) and the empirical nature of our analysis. Given a larger rate sample size, we would expect that confidence interval to significantly narrow.

**Comment 4:** Line 408: Should '(Tanioka and Matsumoto, 2020)' be 'Tanioka and Matsumoto (2020)'?

Thank you for pointing out this error. We corrected this citation format in the revised version.

**References**

Beutel, M. and Wilson, D.: Targeted Oxygen Addition to Hood Canal: A Potential Management Strategy to Ameliorate the Impacts of Hypoxia, in: Proceedings of the 2005 Puget Sound Georgia Basin Research Conference, Puget Sound Georgia Basin Research Conference, Seattle, Washington, 2005.

Stigebrandt, A., Liljebladh, B., de Brabandere, L., Forth, M., Granmo, Å., Hall, P., Hammar, J., Hansson, D., Kononets, M., Magnusson, M., Norén, F., Rahm, L., Treusch, A. H., and Viktorsson, L.: An Experiment with Forced Oxygenation of the Deepwater of the Anoxic By Fjord, Western Sweden, AMBIO, 44, 42–54, https://doi.org/10.1007/s13280-014-0524-9, 2015.

Wallace, D. W. R., Jutras, M., Nesbitt, W. A., Donaldson, A., and Tanhua, T.: Can green hydrogen production be used to mitigate ocean deoxygenation? A scenario from the Gulf of St. Lawrence, Mitig Adapt Strateg Glob Change, 28, 56, https://doi.org/10.1007/s11027-023-10094-1, 2023.

---

## Author Comment (AC3)

Dear Reviewer,

We would like to extend our thanks for your constructive review of our manuscript. Below, we respond point by point to each of your comments.

**Format and Structure**

**Comment 1:** The formatting between sections 1 and 1.1 is unusual and should be revised for consistency.

To improve consistency, we have revised the structure of Section 1 to include subheadings for all content. Specifically, the general introduction text has now been incorporated into "1.1 Background", and the subsequent sections have been re-numbered to "1.2 Previous modeling studies" and "1.3 Research objectives".

**Comment 2:** Sections 3.4, 3.5, and the last paragraph of the Conclusion could be revised into a dedicated Discussion section. Currently, the discussion is fragmented and lacks cohesion. A clear and standalone discussion would improve the paper's overall impact.

The authors had discussed and agreed on combining results and discussions because the paper deals with 5 thematically distinct yet related topics. For each, the results are presented alongside their interpretation to maintain a logical narrative. We recognise that there are both advantages and disadvantages to this structure but decided that, in this case, it was better to combine results and discussion. We note that Reviewer 2 commented positively on the clarity of the paper's logic which we feel would be impacted negatively if we were to return to a more conventional structure. Therefore, while respecting the reviewer's comment, we prefer to keep the structure of the paper as is.

**Comment 3:** Consider incorporating a paragraph that synthesizes past findings and highlights how this study builds on previous work. This could serve as a conceptual model-like summary and strengthen the narrative arc from historical data analysis to future outlook.

We would like to note that a synthesis outlining how this study builds on past work is already integrated within Section 1.2 (now 1.3). A detailed review and comparison of previous modeling efforts, including differences in model structure, coefficients, and OUR is already provided in Supplementary Material Section S1. This section is explicitly referenced in Section 1.1 (now 1.2) of the manuscript to guide readers seeking further background and context.

**Purpose and Abstract**

**Comment 4:** While the final paragraph of the paper presents a clearer articulation of the study's aim, this clarity is not reflected in the Abstract. The Abstract currently focuses too much on methods and lacks a high-level synthesis of the findings and implications. Revise the Abstract to emphasize better the study's scientific significance, including the conceptual contribution and potential applications.

We revised the Abstract to more clearly highlight the study's scientific significance and broader implications. We believe that the updated Abstract now better reflects the scope, findings, and applied relevance of the manuscript.

**Discussion**

**Comment 5:** Some explanations—for example, differences in coefficients used in prior studies—are mentioned but not thoroughly discussed. Please elaborate on the potential consequences of these coefficient differences on the model output or interpretation.

We agree that transport parameter assumptions can significantly affect modeled biogeochemical rates. In response, we added clarification to the final paragraph of Section 3.3 to emphasize that the choice of transport parameters has a strong influence on rate estimates. In our along-channel framework, faster

advection or greater diffusivity reduces residence time which, in turn, requires higher remineralization rates to reproduce the observed along-channel concentration gradients. We now explicitly state that this sensitivity helps explain the wide range of OUR estimates in the literature and show how recalculating previous studies using the tracer-constrained advection speed from TReX ($5 \times 10^{-3}$ m s$^{-1}$) yields rates more consistent with our findings.

**Comment 6:** A more comprehensive discussion could highlight the novelty of this long-term dataset, its value for understanding biogeochemical cycling, and its implications for future monitoring or management.

We completely agree that the long-term dataset is a core strength of this study. To highlight this, we added a sentence in the Conclusion to emphasize its uniqueness and relevance to understanding long-term change in the Gulf and St. Lawrence Estuary. We would like to note that this dataset is only a subset of a larger, integrated, biogeochemical time series spanning the St. Lawrence Estuary, Gulf, and Saguenay Fjord. This full data-product, that includes 20 years of oxygen, carbon, nutrient, transient tracer, and stable isotope measurements is archived through the Canadian Integrated Ocean Observing System – St. Lawrence Global Observatory (CIOOS-SLGO) and will be formally released following the submission of a descriptive/instructive companion data paper to *Earth System Science Data* later this year. The full scope and significance of the data product will be described in greater detail in that publication.

**Comment 7:** Line 424. This appears to be the key assumption of the entire study. Could the authors comment on any potential side effects or unintended consequences associated with this assumption?

We presume the reviewer is referring to the assumption that pure oxygen from industrial sources could be injected and retained within the deep inflow at Cabot Strait. This assumption and its limitations are addressed in the revised manuscript in the final paragraph of Section 3.5. There, we detail the simplifications of our model, including the assumed 100% oxygen retention and the preservation of the water column stratification. We also clarify that the goal of this study is to provide a first-order conceptual estimate of re-oxygenation potential. Finally, we state that a more complete evaluation of the remediation strategy (including injection efficiency, plume dynamics, and system feedback) would require a 3D framework.

**Comment 8:** Model Limitations are mentioned in lines 493-497. Although the authors include error estimates, the limitations of the numerical model are not clearly discussed. Given that short-term variations (e.g., tidal or diurnal changes) may exceed long-term trends in certain locations, the authors should discuss how these short-term dynamics may impact model accuracy and interpretation.

We agree that short-term and spatially localized dynamics, such as tidal mixing and slope-driven variability, can influence dissolved metabolite distributions in ways that are not captured by a 1D model. We added a couple of paragraphs to Section 3.2 that provide estimates of potential vertical DO fluxes (using Fick's Law) under a range of plausible diffusivity scenarios. We compare these values to the modeled OUR and demonstrate that, even under elevated mixing assumptions, vertical diffusion is a secondary contributor to the DO budget on a regional scale. We also noted that localized mixing and cross-channel transport may redistribute metabolites laterally, and that these effects cannot be resolved in our 1D framework. This limitation underscores the need for more comprehensive 3D modeling approaches to fully evaluate short-term and spatially variable dynamics.

**Figures**

**Comment 9:** Many figures are overly crowded. For instance, Y-axis labels are dense, and contour lines frequently overlap with annotations.

We carefully examined all figures and chose to revise Figures 3, 5, and 6 to improve readability. For figure 3, contour labels were decreased in font size and frequency to avoid overlap while contour lines were thinned out and smoothed. For figures 5 and 6, the y-axis was decreased to count in 2-year increments and the x-axis in 100-km increments to decrease clutter. We believe these changes improve visual clarity without compromising the presentation of key information. The revised figures are presented below.

[Figure]

**Revised Figure 3**

[Figure]

**Revised Figure 5**

[Figure]

**Revised Figure 6**